# THREAD: A LOGIC-BASED DATA ORGANIZATION PARADIGM FOR HOW-TO QUESTION ANSWERING WITH RETRIEVAL AUGMENTED GENERATION

## ABSTRACT

Recent advances in retrieval-augmented generation have significantly improved the performance of question-answering systems, particularly on factoid '5Ws' questions. However, these systems still face substantial challenges when addressing '1H' questions, specifically how-to questions, which are integral to decision-making processes and require dynamic, step-by-step answers. The key limitation lies in the prevalent data organization paradigm, chunk, which divides documents into fixed-size segments, and disrupts the logical coherence and connections within the context. To overcome this, in this paper, we propose THREAD, a novel data organization paradigm aimed at enabling current systems to handle how-to questions more effectively. Specifically, we introduce a new knowledge granularity, termed 'logic unit', where documents are transformed into more structured and loosely interconnected logic units with large language models. Extensive experiments conducted across both open-domain and industrial settings demonstrate that THREAD outperforms existing paradigms significantly, improving the success rate of handling how-to questions by 21% to 33%. Moreover, THREAD exhibits high adaptability in processing various document formats, drastically reducing the candidate quantity in the knowledge base and minimizing the required information to one-fourth compared with chunk, optimizing both efficiency and effectiveness.

## 1 INTRODUCTION

Question answering (QA) is a foundational research topic in human-machine interaction (Allam & Haggag, 2012). Among the most advanced techniques, Retrieval-augmented generation (RAG) organizes external documents into fixed-size chunks and retrieves relevant knowledge to enhance QA systems (Shao et al., 2023; Trivedi et al., 2023; Jiang et al., 2023; Asai et al., 2023). This routine is particularly effective in handling the '5Ws' questions, which are typically factoid requiring factual information about specific details (Yang et al., 2018; Jiang et al., 2019; Kwiatkowski et al., 2019; Stelmakh et al., 2022), such as *'When is Shakespeare's birthday?'*. Specifically, the data organization paradigm only needs to provide chunks containing the relevant knowledge, e.g. triples or documents about the topic entity, to the RAG systems to facilitate the '5Ws' questions. However, the '1H' questions, derived from Aristotle's Nicomachean Ethics (in this paper as 'how-to' questions, Crisp 2014), remain largely underexplored[1]. These questions are in high demand in practical applications such as teaching us how to write code to achieve specific goals.

Central to problem-solving (Polya & Pólya, 2014) and human learning in cognitive science (Learn, 2000), how-to questions are inherently more complex, often involve processes that require interpretation and analysis (Deng et al., 2023b). For example, correctly answering the question shown in Figure 1 *'How to diagnose and fix a performance issue in a web application?'* involves a multi-step decision-making process, i.e., firstly checking the server load and response time, then optimizing server configuration or indexes and queries with user feedback. Such a dynamic nature necessitates the RAG systems incrementally guide users through each step, adapting to specific circumstances and providing precise and logical information. However, the prevalent data organization paradigm,

---

[1]The '5Ws' represent What, Why, When, Where, and Who, and the '1H' stands for How. The 'Why' question can sometimes delve into explanations and motivations.

Figure 1: An example of how-to questions with its decision-making process. We omit details such as the actions to check the server load and response time due to limited space.

chunk[2] (Splitter, 2023; Chen et al., 2023; Gao et al., 2023) leaves current RAG systems (Asai et al., 2023; Shao et al., 2023) struggling to solve such questions. This chunk-based paradigm disrupts the logical flow and coherence of the content due to its inability to effectively represent the internal logic. Consequently, it returns excessive information during inference, making it difficult to maintain continuity between sequential steps, leading to fragmented responses for how-to questions and posing significant challenges for users. Overcoming these limitations necessitates a paradigm shift that accounts for the logical structure and stepwise nature of how-to questions, enabling more precise and coherent information retrieval tailored to specific needs.

In this paper, we propose THREAD, a new logic-based data organization paradigm designed to enhance the handling of how-to questions. The name THREAD evokes the idea of '*Pulling on the **thread**, the whole mystery started to unravel like a sweater.*' (Garcia & Stohl, 2011). Specifically, we introduce a new knowledge granularity named 'logic unit', comprising five key components and four different types (see §3.1 and §3.2). We employ a two-stage process depicted in §3.3 to extract logic units (LUs) from documents. The first, optional stage is reformulating the original documents depending on their format and style, and the second focuses on extracting and merging logic units. In this way, THREAD captures connections within the documents, breaking them into more structured and loosely interconnected logic units. When answering how-to questions, the system integrated with THREAD enables a dynamic interaction manner. First, it retrieves relevant LUs based on their indexed headers. Then, the body of the selected LU provides the necessary content to generate responses for the current step. With user feedback, the linker in LU dynamically connects to other LUs, allowing the system to adapt its responses until the how-to question is comprehensively addressed.

We evaluate the effectiveness of THREAD by conducting experiments in two open-domain, web navigation (Deng et al., 2023a) and Wikipedia Instructions (Koupaee & Wang, 2018), and one industrial setting, Incident Mitigation (Shetty et al., 2022). Experimental results demonstrate that while existing paradigms struggle with how-to questions, THREAD excels at handling consecutive steps and consistently outperforms them, especially in real-world incident mitigation scenarios, with an improvement of Success Rate ranging from 21.05% to 33.33%. In addition, THREAD demonstrates great efficiency and effectiveness in processing various document formats, significantly reducing the number of retrieval units to 20% and the length of tokens needed for generation to 22.60%. The main contributions of this paper include:

- We highlight the challenges faced by current RAG systems in addressing how-to questions. To address the limitation of the chunk-based paradigm, we propose THREAD, a novel data organization paradigm that transforms original documents into structured, interconnected logic units.

- Integrated with THREAD, our system follows a dynamic interaction mode, guiding users incrementally through each step and adapting to their specific circumstances. Our system also brings more possibilities for an automation pipeline, solving how-to questions more efficiently.

- Experimental results show that THREAD significantly outperforms existing data organization paradigms across three scenarios. Additionally, THREAD requires less information, while effectively handling various document formats and substantially reducing the knowledge base size.

---

[2]'Chunk' refers to a general document splitting paradigm including chunks, sentences, phrases, etc.

## 2    RELATED WORK

**Data Organization Paradigm in RAG**    The data organization process is a critical pre-stage of RAG methods where documents are processed and segmented following certain data organization paradigms. The most common data organization paradigm is splitting documents into retrieval units (Gao et al., 2023). These retrieval units vary in granularity such as phrases, sentences, propositions (Chen et al., 2023), chunks (Kamradt, 2024), etc. Coarser-grained units contain more information but introduce redundant noise, while finer-grained units have lower semantic integrity and often require retrieving more units to gather comprehensive information. However, the chunk-based data organization paradigm ignores the logical and relational connections between chunks, potentially disrupting the inherent logic flow in documents. Another paradigm constructs documents into knowledge graphs (KG), where retrieval units include entities, triplets, etc. (Gaur et al., 2022; Sen et al., 2023; He et al., 2024; Wang et al., 2024; Edge et al., 2024). These approaches emphasize semantic/lexical similarities between retrieval units, their success in factoid questions is limited when applied to how-to questions. This limitation arises because how-to questions demand logical connections between retrieval units that extend beyond mere semantic or lexical similarities.

**Information Retrieved by RAG**    The effectiveness of RAG methods depends on the generator's ability to utilize retrieved information and the quality and quantity of that information. Insufficient question-relevant information can cause hallucinations in LLM-based generators (Li et al., 2023; Zhang et al., 2023), making it crucial to improve the retrieval process. Traditional one-round retrieval methods (Guu et al., 2020; Lewis et al., 2020) often fail to gather all necessary information due to their reliance on the similarity between query and retrieval units (Gan et al., 2024). Advanced RAG methods use query rewriting and expansion (Shao et al., 2023; Trivedi et al., 2023; Kim et al., 2023) or iterative retrieval (Shao et al., 2023; Jiang et al., 2023; Asai et al., 2023) to collect more information. However, these approaches still struggle with how-to questions, which require making next-step decisions based on the current retrieved units, unless the current retrieved units contain clues that lead to the next step. The main issue is the lack of connections between retrieval units, which prevents effective retrieval and the gathering of sufficient information.

## 3    METHODOLOGY

In this section, we introduce 'logic unit' with its components and types, then explain the process of constructing the THREAD knowledge base, followed by an illustration of how THREAD integrates with existing systems.

### 3.1    LOGIC UNIT: RETRIEVAL UNIT OF THREAD

We propose a new knowledge granularity called 'logic unit'. Different from the chunk-based paradigm, LU comprises specially designed components, especially for bridging consecutive steps when addressing how-to questions, considering the internal logic and coherence of documents.

*Prerequisite.* The prerequisite component acts as an *information supplement*, providing the necessary context to understand the LU. For example, an LU may include domain-specific terminology such as entities or abbreviations. The prerequisite explains these terms and can generate new queries to retrieve LUs with more detailed information. Without this context, passing these LUs to an LLM-based generator could lead to hallucinations. Additionally, the prerequisite can function as an *LU filter*, containing constraints that must be met before the LU is considered in answer generation. This filtering ensures only relevant LUs are retrieved. As shown in step LU in Figure 2, accessing the server monitor is a prerequisite for checking the server load.

*Header.* The header summarizes the LU or describes the intention it aims to address, depending on the type of LU (refer to §3.2). For example, the header could be the name of a terminology if LU describes a terminology; if the LU describes actions to resolve a problem, the header describes the intent or the problem LU aims to resolve. Different from chunk that indexes the entire content, we use the header for indexing which serves as the key for retrieving the LU based on a query.

*Body.* The body contains detailed information on the LU, which is the core content fed into the LLM-based generator to generate answers. It includes specific actions or necessary information such

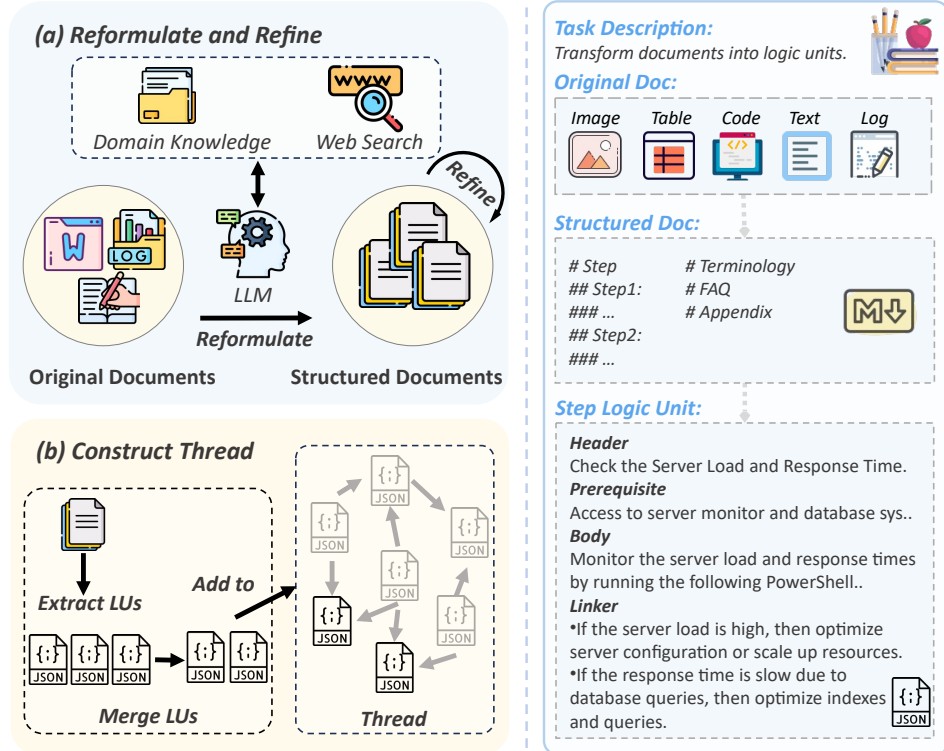

Figure 2: The construction process of THREAD. The left part depicts the two-stage process including reformulating documents into structured ones and extracting and merging logic units, the right part instantiates this process with a concise example of a document and its corresponding extracted LU.

as code blocks, detailed instructions, etc. This detailed content helps resolve the query mentioned in the header or provides a detailed explanation of the header.

***Linker.*** The linker acts as a bridge between logic units, enabling the dynamic process of how-to questions. Unlike the chunk-based paradigm, which relies on previous retrieval units that often lack direct clues, the linker in THREAD provides necessary information to generate new queries for subsequent retrieval. In Figure 2, the linker of 'checking server' specifies multiple possibilities after taking the action in LU body, guiding the retrieval of the next-step LU. Its format varies by LU type, serving as either a query for retrieving other LUs or an entity relationship. The edge of knowledge graph in traditional factoid questions is a special linker enabling navigation between related entities. When no further LUs are connected, the linker remains empty, isolating the current LU.

***Meta Data.*** The meta data includes information about the source document from which the LU is extracted, such as the document title, ID, date, and other relevant details. This meta data is crucial for updating LUs when the source documentation is revised and reprocessed.

## 3.2 LOGIC UNIT TYPE

When converting documents into logic units, we distinguish how-to questions into *'linear'* and *'dynamic'* types, where linear how-to questions often involve a fixed sequence of steps that do not require feedback or decision points based on intermediate outcomes. Below are the common LU types identified in our experiments[3], showing THREAD is versatile enough to handle diverse types:

***Step.*** This is the most common LU type for resolving how-to questions. Each LU body represents detailed actions, including code blocks and resolution instructions. The LU prerequisite describes the actions that need to be completed before executing the current actions. The prerequisite is especially crucial for identifying the entry point of a solution. For example, when facing a problem, there exist

---

[3]These LU types are summarized from our practice in experiments with industrial and public datasets. There may be additional LU types depending on the specific scenario.

different ways to resolve it depending on the current situation. The prerequisite serves as a condition in the LU selection stage, filtering out LUs that do not meet the prerequisite.

***Terminology.*** This type provides detailed explanations of domain-specific terminology. For example, terms may share the same name or abbreviation in the LU header but convey different meanings in the LU body. The prerequisites for terminology LUs describe scenarios where the terminology typically appears and linkers are usually empty unless referencing extended terminology that depends on it.

***FAQ.*** This type provides frequently asked questions, supplementing the knowledge base. These LUs are typically isolated, with the LU body offering solutions through sequential steps that address linear how-to questions not reliant on dynamic states. They save time by avoiding the need for sequentially retrieving LUs for common questions.

***Appendix.*** This type provides additional information relevant to the scenario of LUs, such as examples, background, lookup tables, etc. These LUs serve as supplementary knowledge for LLMs when generating responses or executable plans.

### 3.3 THREAD: LU-BASED KNOWLEDGE BASE

In practice, documentation is often unstructured and varies in format and style. Our approach to converting documentation into THREAD involves a two-stage process to obtain LUs, as shown in the left part of Figure 2. We believe that with advancements in LLMs, this process could be streamlined into a single stage, enabling high-fidelity LU conversion with minimal hallucination.

***Documentation Reformulation (Optional).*** This stage is optional, depending on the quality of the documentation. For example, in software engineering, Troubleshooting Guides (TSGs) often vary in format, include diverse types of information, and lack readability and detail, negatively impacting productivity and service health (Shetty et al., 2022). Due to such format, where some are clearly outlined and others are disordered, we avoid directly extracting LUs from original documents. Instead, we first reformulate these documents into structured formats. Leveraging LLMs for this task, we enhance the LLMs' in-domain understanding by providing search capabilities and domain-specific context. This is followed by a refinement step to prevent overlooking details or hallucinating information. Figure 2 (a) shows the reformulation stage and the right represents with symbols. But it is unnecessary for well-written documents like product help docs, which typically follow a linear how-to format. We illustrate the process in Table 9 by example and provide prompts in Appendix C.1.

***LU Extraction and Merge.*** After reformulation, multiple LUs of varying types can be extracted from a single structured document (shown in Figure 2 (b)). Unlike chunk-based data organization commonly with fixed chunk sizes, LU granularity depends on content. For example, solutions to linear how-to questions typically form a single path from start to completion, with interconnected steps and no multiple execution outcomes encapsulated in one LU, such as an FAQ LU. However, for dynamic how-to questions with multiple possible outcomes, it is better to have one step per LU (Step LU), with Linkers navigating to the next LUs. Note that in dynamic how-to questions, not every step has multiple execution outcomes. If only one next step exists, the LUs can be merged to include both current and subsequent steps. Additionally, LUs with similar Headers and Bodies should be merged, extending the Prerequisite and Linker. More details about merging are provided in Appendix C.2.

***LU update.*** In industry, documentation is often updated with each product version release. When this happens, we redo the above steps for the updated documentation, identifying LUs in THREAD with their Meta Data and replacing outdated LUs.

As we extract and merge LUs, the collection of LUs from all documents forms the knowledge base, THREAD. This LU-based knowledge base serves as an essential component compatible with the current RAG system and even makes it possible for automation.

### 3.4 INTEGRATE THREAD WITH QA SYSTEM

To demonstrate how THREAD works, we use dynamic how-to questions as an example. Figure 3 shows the RAG system incorporating our THREAD data organization paradigm. The *Retriever* and *LLM-based Generator* are inherited from the original RAG system. LUs are indexed by their Headers. When an initial how-to question is submitted, the Retriever identifies the top-K most relevant LUs based on query-header similarity. The Selector then checks the prerequisites of these LUs and filters

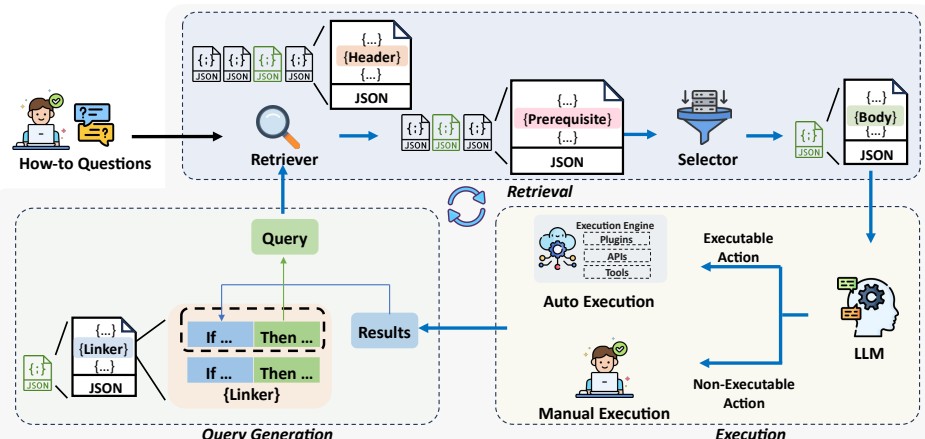

Figure 3: The RAG system integrated with THREAD. It retrieves relevant LUs based on query-Header similarity and filters out LUs that do not meet the current Prerequisites. The selected LUs are passed to LLMs to generate actions based on Body for execution. After execution, the Linker matches results and generates a new query for the next retrieval iteration.

out those that do not meet the current prerequisites, derived from the initial question or any available chat history. If no current prerequisite is provided, the system can prompt the user, e.g., 'Before doing ..., have you tried ...?', to obtain the current prerequisite for LU filtering. After selection, the body of the LUs is fed into the LLM-based generator to produce an answer. If an execution engine is available, actions can be executed automatically; otherwise, the answer/action is presented to the user for manual execution. Once the action is executed, the Linker matches one of the possible outcomes and generates a new query for the next retrieval round.

Unlike traditional RAG systems, THREAD-enabled systems can be potentially fully or semi-automated when integrated with execution engines. This integration offers greater automation and flexibility, as updating LUs automatically updates the system, compared to manually designed pipelines (see §3.3).

## 4 EXPERIMENTAL SETUP

### 4.1 SCENARIOS AND DATASETS

We evaluate THREAD on two open-domain scenarios: Web Navigation (Mind2Web (Deng et al., 2023a)) and Wikipedia Instructions (WikiHow (Koupaee & Wang, 2018)), and one industrial setting: Incident Mitigation (IcM (Shetty et al., 2022; An et al., 2024)). We provide one example of each dataset in Appendix B to better demonstrate linear and dynamic how-to questions.

*Web Navigation.* Mind2Web (Deng et al., 2023a) is a dataset designed for web agents to perform complex tasks on real-world websites based on language instructions. Each task involves a 'dynamic how-to question', with multiple possible outcomes depending on the state of the executed actions.

*Wikipedia Instructions.* WikiHow[4] is a platform containing numerous articles that offer step-by-step guidance on various procedural tasks. Each article is typically titled 'How to' and includes a brief task description, followed by a linear, fixed sequence of steps.

*Incident Mitigation.* Incident Mitigation (Shetty et al., 2022; An et al., 2024; Jiang et al., 2024) is essential for operating large-scale cloud services, where engineers use Troubleshooting Guides (TSGs) to address incidents. Each step in incident mitigation can yield different outcomes depending on the state, making it suitable for testing THREAD on 'dynamic how-to questions'. Unlike the other open-domain datasets, we perform a human evaluation involving twenty on-call engineers (OCEs) responsible for incident mitigation. We collect five incidents, classified into two simple and three hard ones based on their mitigation steps in history. Each OCE is tasked with mitigating all five incidents, using randomly ordered baselines per incident to avoid familiarity bias. The RAG system initiates

---

[4] https://www.wikihow.com

| Dataset | #Docs | #Tasks | #Steps | #Chunks | #LUs | Dynamic | Executable |
|---------|-------|--------|--------|---------|------|---------|------------|
| Mind2Web | 490 | 252 | 2094 | 6210 | 1089 | ✓ | ✓ |
| WikiHow | 97 | 97 | 2140 | 4225 | 774 | ✗ | ✗ |
| IcM | 56 | 95 | 323 | 413 | 378 | ✓ | ✓ |

Table 1: The statistics and characteristics of datasets, including the number of documents, LUs, etc.

automated mitigation for each incident; if it encounters a failure at any step, an OCE intervenes to address the issue before the system resumes automated procedures[5].

## 4.2 DOCUMENTS FOR RETRIEVAL

As the Mind2Web dataset lacks relevant documents, we create retrieval documents tailored to the RAG system. Assuming that each website has help docs applicable across different tasks, we use the "Cross-Task" test set, selecting examples from the training set to craft informative documents for each website. And we follow Wang et al. (2023) to brainstorm different formats of documents. More details about collecting documents are shown in Appendix C.2.

For the WikiHow dataset, we utilize publicly available Windows Office Support Docs[6] as retrieval documents. We select around 100 tasks from WikiHow tagged with Microsoft products like Word, PowerPoint, and Teams, each containing 10 to 40 steps related to Windows operations.

For the IcM dataset, we collect 56 TSGs from an enterprise-level engineering team responsible for a large-scale cloud platform. The selected incidents in the IcM dataset can be resolved using the knowledge provided in these TSGs. We list the statistics of our datasets in Table 1.

## 4.3 BASELINES

In the Mind2Web dataset, previous work has not treated it as how-to questions. State-of-the-art methods like SYNASE (Zheng et al., 2023) and MINDACT (Deng et al., 2023a) either use In-Context Learning (ICL), providing few-shot demonstrations, or Supervised Learning (SL) to finetune a model on the training set. To ensure a fair comparison with our LLM endpoints[7], we re-implement the MINDACT experiments with the same demonstrations and included chat history as extra context. In our paper, we treat the Mind2Web dataset as dynamic how-to questions and use our RAG system to solve these tasks. For comparison, we use doc-based (providing the entire document) and chunk-based data organization paradigms as baselines against THREAD. This RAG system with different data organization paradigms also serves as baselines for the WikiHow and IcM datasets. More details about experiments can be found in Appendix C, D.

## 4.4 EVALUATION METRICS

For the Mind2Web dataset, each task is treated as a multi-choice question. At each step, the input consists of HTML code, an instruction, and a set of choices, while the output is the selected choice, operation, and an optional value. We adapt the evaluation metrics from (Deng et al., 2023a), which include: *Element Accuracy (Ele. Acc)* to evaluate the chosen HTML element; *Operation F1 (Op. F1)* to calculate the token-level F1 score for predicted operations such as "CLICK", "TYPE IN", etc.; *Step Success Rate (Step SR)*, where a step is successful if both the selected element and predicted operation are accurate; and *Success Rate (SR)*, where a task is successful only if all steps are successful.

For the WikiHow dataset, which contains ground truth steps, we leverage LLMs to extract "Action Items" from each ground truth step and generated step, and we use the following metrics: *Precision* ($P = \frac{\#matched\_items}{\#total\_generated\_items}$); Recall ($R = \frac{\#matched\_items}{\#total\_groundtruth\_items}$); *F1*; and *Success Rate (SR)* to assess if the generated steps can successfully complete the task, using LLMs to evaluate (Appendix C.3 shows the evaluation prompt).

---

[5]Varying across new-hire and experienced OCEs. Mitigating one incident costs each OCE around 30 minutes to 1 hour manually, and we collect five incidents to ensure consistency, engagement, and ethical treatment. Note that one OCE's data was contaminated during the experiment, so we removed that OCE's data.

[6]https://github.com/MicrosoftDocs/OfficeDocs-Support

[7]We use GPT-3.5 and GPT-4 with version 1106-preview.

| Method | Model | Paradigm | Mind2Web Cross-Task | | | |
|---|---|---|---|---|---|---|
| | | | Ele. Acc | Op. F1 | Step SR | SR |
| SYNASE (2023) | w/ GPT-3.5* | ICL | 34.00 | - | 30.60 | 2.40 |
| MINDACT (2023a) | w/ GPT-3.5 | ICL | 40.69 | 49.66 | 33.91 | 1.59 |
| | w/GPT-4 | ICL | 62.80 | 60.37 | 51.81 | 10.32 |
| | w/ Flan-T5$_{XL}$* | SL | 55.10 | **75.70** | 52.00 | 5.20 |
| RAG | w/ GPT-4 | Chunk | 64.23 | 65.96 | 58.45 | 8.73 |
| | w/ GPT-4 | Doc | 63.80 | 65.89 | 58.36 | 11.51 |
| | w/ GPT-4 | THREAD | **68.29** | 69.53 | **61.94** | **12.30** |

Table 2: Experiment results on Mind2Web. '*' represents taking results from the original paper.

| Paradigm | Category | Incident Mitigation | | | | |
|---|---|---|---|---|---|---|
| | | SR | Step SR | P.F. Step SR | HI | Turns |
| Chunk | Simple | 40.51 | 60.90 | 60.90 | 30.10 | 3.14 |
| | Hard | 28.95 | 53.16 | 43.05 | 46.84 | 6.84 |
| Doc | Simple | 43.86 | 63.90 | 63.90 | 36.09 | 2.98 |
| | Hard | 31.58 | 57.89 | 42.11 | 42.11 | 6.53 |
| THREAD | Simple | **77.19** | **88.72** | **84.21** | **11.28** | **2.56** |
| | Hard | **52.63** | **84.21** | **68.95** | **15.79** | **5.74** |

Table 3: Experiment results on Incident Mitigation. We divide incidents into two groups: simple or hard, and compare the performance separately.

For the IcM dataset, which involves task execution, we perform evaluations with OCEs (refer to §4.1) using five metrics: *Success Rate (SR)* indicating the percentage of incidents mitigated automatically by the system without human intervention; *Step Success Rate (Step SR)* representing the percentage of successful steps out of all task steps; *Pre-Failure Step Success Rate (P.F. Step SR)* representing the percentage of successful steps before the first failure; *Human Intervention (HI)* measuring the percentage of steps requiring human intervention; and *Average Turns (Turns)* to measure the average interaction turns between OCEs and the system during incident mitigation.

## 5 EXPERIMENTAL RESULTS

### 5.1 MAIN RESULTS

***Web Navigation.*** The overall performance on Mind2Web is shown in Table 2, where we compare our method with baselines and both doc-based and chunk-based RAG methods. We observe that providing informative documents (regardless of the data organization paradigm) within RAG methods significantly improves performance. RAG methods outperform ICL and SL methods. By incorporating external documents, the doc-based RAG method achieves performance comparable to the best results of MINDACT. Notably, our THREAD paradigm is the best among RAG methods, showing improvements of 4.06% in Ele. Acc, 3.49% in Step SR and 3.57% in SR. Note that MINDACT-SL gets the highest Op. F1 due to label distribution imbalance[8], leading the model to favor generating the most frequent operations.

***Incident Mitigation.*** Table 3 illustrates the advantages of THREAD over other paradigms when addressing complex dynamic how-to questions. Both chunk-based and doc-based RAG methods exhibit significant limitations in incident mitigation, as their SR and P.F. Step SR are much lower due to their inability to connect subsequent steps based on the current step. In contrast, our THREAD achieves the best score across all metrics, particularly achieving a significant increase in SR from 21.02% to 33.33%. More importantly, the highest *P.F. Step SR* highlights THREAD's ability to dynamically connect to subsequent steps based on user feedback, thereby reducing the need for human intervention. Consequently, THREAD not only achieves the highest Step SR but also requires the fewest interaction turns to effectively mitigate both simple and complex incidents.

---

[8]If the model predicts all operations as "CLICK," the Op. F1 would reach 79.90%.

*Wikipedia Instructions.* Table 4 presents the performance of various data organization paradigms on WikiHow tasks, experimenting with both single-turn and multi-turn interactions. In the single-turn setting, where the RAG system needs to generate the entire plan in a single interaction, our THREAD outperforms the doc-based method with 5.15%. This indicates that THREAD delivers information with less redundancy while achieving a higher SR score. Additionally, when comparing THREAD to the chunk-based method, we observe a significantly lower success rate (SR) of 19.59%. This result highlights the challenge of retrieving all relevant chunks without considering the logic flow inside documents.

| Paradigm | WikiHow | | | |
|---|---|---|---|---|
| | SR | P | R | F1 |
| **Single-Turn** | | | | |
| Chunk | 19.59 | 60.20 | 25.16 | 35.49 |
| Doc | 58.76 | 77.71 | 57.93 | 66.37 |
| THREAD | **63.91** | **83.43** | **71.48** | **76.99** |
| **Multi-Turn** | | | | |
| Chunk | 20.62 | 52.95 | 25.54 | 34.45 |
| Doc | 68.04 | 87.10 | 70.65 | 78.02 |
| THREAD | **72.16** | **89.77** | **73.36** | **80.74** |

Table 4: Experiment results on WikiHow with different interaction manners and paradigms.

In the multi-turn setting, the RAG system performs iterative retrieval, showing superior performance compared to single-turn. This indicates the advantage of a step-by-step approach in handling how-to questions. We can see that the SR improves significantly from 58.76% to 68.04% in the doc-based paradigm and from 63.91% to 72.16% in THREAD. Similar to the single-turn setting, the chunk-based method splits documents while disrupting internal logic, resulting in a low SR of 20.62%. Our THREAD excels across all metrics, achieving the highest SR of 72.16%, highlighting its effectiveness in maintaining and modeling the connections between steps.

## 5.2 ABLATION ON RAG SYSTEM SETTINGS

This section presents an ablation study on the key settings of our RAG system. While retriever and generator variants have been explored in other research (Gao et al., 2023), we use text-embedding-ada-002 (OpenAI, 2022) for the retriever and GPT-4 as the LLM-based generator. The ablation study is performed on Mind2Web dataset.

*Multi-turn Interaction.* As shown in Table 4, the multi-turn setting is better than the single-turn setting in answering how-to questions. We use multi-turn setting for the other scenarios in this paper.

*Chat History.* The chat history helps the system to check previous actions and results, helping the system make better decisions. Without chat history, the performance drops as shown in Table 5. We include chat history for all RAG-based methods in our experiments.

*Retrieval Units Selector.* As aforementioned in §3.4 and Figure 3, the Selector picks the most relevant retrieval units from the top-K retrieved units. Table 5 shows the ablation of the retrieval unit selector. Comparing chunk with and without selector, the performance of Ele. Acc drops by 4.29% and Op.F1 by 3.38%. Conversely, the selector improves all metrics with applied to THREAD. Unlike LU selection, which filters out irrelevant LUs based on prerequisites, chunk selection disregards inter-chunk connections and

| Paradigm | Ele. Acc | Op. F1 | Step SR |
|---|---|---|---|
| ICL | **62.80** | **60.37** | **51.81** |
| *w/o.* historical steps | 56.97 | 59.09 | 50.38 |
| Chunk | 59.94 | 62.58 | 54.39 |
| *w/o.* chunk selection | **64.23** | **65.96** | **58.45** |
| THREAD | **68.29** | **69.53** | **61.94** |
| *w/o.* LU selection | 67.05 | 68.43 | 60.79 |

Table 5: Ablation study of integrating chat history and retrieval unit selection on Mind2web.

may exclude relevant chunks. Therefore, we activate the selector only for our THREAD.

## 5.3 ANALYSIS OF DATA ORGANIZATION PARADIGMS

We compare various data organization paradigms within RAG, implementing Semantic Chunking (Kamradt, 2024) and Proposition (Chen et al., 2023) in addition to recursive chunking (Splitter, 2023) (chunk-based) and entire document (doc-based) approaches (details in Appendix A). Table 6 shows that our THREAD paradigm outperforms the other paradigms across all metrics. Although Semantic and Proposition use LLMs to merge semantically similar sentences, they fail to adequately address logical connections. Additionally, our system processes the smallest token length of retrieval units yet achieves the highest performance, underscoring our approach's efficiency and effectiveness[9].

---

[9]We analyze the cost and scalability of THREAD in Appendix E.

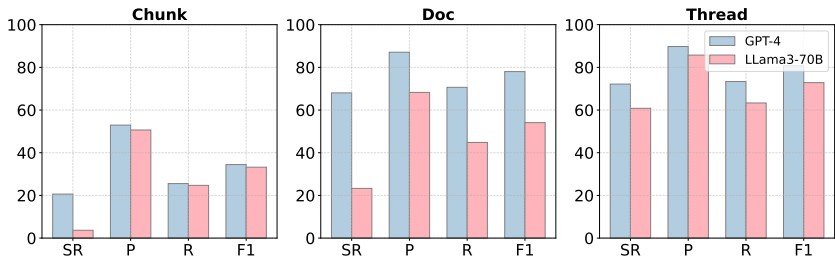

Figure 4: Analysis of using different LLMs on WikiHow (GPT-4 v.s. LLaMA3-70B).

| Paradigm | Ele. Acc | Op. F1 | Step SR | #Tokens in RU |
|---|---|---|---|---|
| Doc | 63.80 | 65.89 | 58.36 | 663.84 |
| Recursive | 64.23 | 65.96 | 58.45 | 695.77 |
| Semantic | 65.14 | 67.30 | 59.93 | 1337.16 |
| Proposition | 62.37 | 64.78 | 56.78 | 790.14 |
| THREAD w/o. | 67.05 | 68.43 | 60.79 | 772.67 |
| THREAD | **68.29** | **69.53** | **61.94** | **157.10** |

| Format | Paradigm | Ele. Acc | Op. F1 | Step SR |
|---|---|---|---|---|
| Structured | Chunk | 64.23 | 65.96 | 58.45 |
| | THREAD | **68.29** | **69.53** | **61.94** |
| Hierarchical | Chunk | 60.60 | 63.46 | 55.06 |
| | THREAD | 66.57 | 67.89 | 60.08 |
| Tabular | Chunk | 56.30 | 59.26 | 51.43 |
| | THREAD | 65.71 | 67.69 | 59.55 |
| Narrative | Chunk | 56.63 | 60.39 | 51.66 |
| | THREAD | 66.24 | 68.22 | 60.17 |

Table 6: Analysis of data organization paradigms.   Table 7: Analysis of different document formats.

## 5.4 DOCUMENT FORMATS IN LU EXTRACTION

As mentioned in §4.2, we test our LU extraction method in accommodating varying document structures. We generate different document formats (detailed in Appendix C.2), including structured markdown, hierarchical guidelines, tabular checklists, and narrative documents. Table 7 shows that our THREAD paradigm effectively organizes these formats, consistently outperforming the chunk-based paradigm across all metrics. Our THREAD improves Ele. Acc by up to 9.61%, Op. F1 by up to 8.43%, and Step SR by up to 8.51%. Especially, our THREAD achieves the highest performance with structured documents, as this format helps construct a higher-quality knowledge base.

## 5.5 DIFFERENT LLMS AS BACKBONE

We further conduct experiments on LLaMA3-70B, as shown in Figure 4. Although LLaMA3-70B is less powerful than GPT-4, it still demonstrates competitive performance with the help of THREAD. Results indicate that LLaMA3 struggles with WikiHow questions applying chunk-based or doc-based paradigms. However, with the integration of THREAD, LLaMA3 not only achieves much better performance in SR, but also narrows the gap with GPT-4. Specifically, LLaMA3 achieves SR and F1 scores of 60.82% and 72.84%, respectively, compared to GPT-4's scores of 72.16% and 80.74%. This indicates that while a performance gap remains, particularly with the chunk-based and doc-based paradigms, our THREAD considerably reduces this disparity between GPT-4 and LLaMA3. These findings highlight the value of our proposed paradigm in enhancing the performance of different LLMs, showcasing its generalizability, robustness, and efficiency in handling how-to questions.

## 6 CONCLUSION

In this paper, we address the overlooked category of handling how-to questions in QA systems by proposing THREAD, a novel data organization paradigm that captures logical connections within documents. By introducing a new knowledge granularity called 'logic unit', THREAD restructures documents into interconnected logic units that are compatible with RAG methods. Extensive experiments show that THREAD significantly outperforms existing paradigms, improving performance while reducing the knowledge base size and minimizing the information needed for generation.

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

## A  CURRENT DATA ORGANIZATION PARADIGM

From Gao et al. (2023), current data organization paradigms can be categorized into phrases, sentences, propositions, chunks, and so on. In our paper, we choose chunks [10] and propositions to compare with our proposed THREAD[11].

**Recursive Chunk.** This chunking method splits the original documents using a list of separators, then reassembles them according to specified chunk sizes and overlap sizes. In our experiment, we use different chunk sizes for each dataset: 1000 for Mind2Web, 2000 for IcM, and 300 for WikiHow. The chunk overlap sizes also vary 50 for Mind2Web, 100 for IcM, and 30 for WikiHow.

**Entire Document.** This method sends the entire document directly into the model, constrained by the document's length and structure.

**Semantic Chunk.** Kamradt (2024) proposes splitting chunks based on semantic similarity. The hypothesis is that semantically similar chunks should be grouped together. By comparing the semantic similarity between adjacent sentences, the method identifies "break points". If the similarity in the embedding space exceeds a certain threshold, it marks the start of a new semantic chunk.

**Agentic Chunk (Proposition).** Chen et al. (2023)[12] introduces the concept of the Proposition Paradigm, which involves extracting independent propositions from original documents. The Agentic Chunk method is based on this paradigm. It first splits the documents into paragraphs, then extracts propositions from each paragraph, and at last merges similar propositions into chunks.

In all our experiments, we set the temperature of LLMs to 0 and top_p to 1 for results reproduction.

## B  EXAMPLE OF DATASET

We list one example of each dataset in Table 8, and an example of the construction process of THREAD including the original document, reformulated document, and its corresponding logic unit in Table 9.

## C  EXPERIMENTAL DETAILS

### C.1  INCIDENT MITIGATION

We take the scenario of incident mitigation to show the instructions about how to construct our knowledge base, including document reformulation, code template extraction, and logic unit selection[13].

> **Instruction that formulates the original unstructured troubleshooting guide into structured one.**
>
> [System]
> You are a helpful troubleshooting guide assistant who helps the user formulate the manual unstructured troubleshooting guide <TSG> into a structured one. The <TSG> is in markdown format, with the first level header describing the incident or problem, and the following second level header providing information related to the incident or problem.
> Each second-level subsection can be categorized into the following types: Terminology, FAQ, STEP, and Appendix. Your reformulation should strictly comply with the following definition:
> - Terminology: firstly, it should be the relationship or connection between terminology about the incident, if not, it can be the explanation or concept of the incident. Sometimes it should be extracted and summarized by yourself.
> - FAQ: frequently asked questions that help to understand the incident.
> - STEP: the processes to resolve the incident, and you should make sure its completeness. Usually, steps have causal inner connection, the former step will trigger the next step.

---

[10]We use the implementation by LangChain https://python.langchain.com/v0.2/

[11]Note: For chunks, we retrieve the top-5 at each time, and for documents, we only retrieve the top-1.

[12]For proposition paradigm, we use agentic chunker since the input token of Flan-T5 is limited to 512, https://github.com/FullStackRetrieval-com/RetrievalTutorials.

[13]The selection strategy is the same for both chunk and LU selections, which leverages LLMs to select the most relevant retrieval element from the retrieved top-K retrieval elements.

| Dataset | Example |
|---|---|
| Mind2web (Dynamic) | ```
<html> ... </html>
Based on the HTML webpage above, try to complete the following task
Task: Book the lowest-priced and quickest flight for 5 adults and 1 child
    on May 20 from Mumbai to any airport near Washington.
Previous actions:
None
What should be the next action?
Please select from the following choices (If the correct action is not in
    the page above, please select A. 'None of the above'):
A. None of the above
B. <div id=0> <input radio triptype roundtrip true /> <label> 
C. <label id=1>  Search flights one way   One
D. <a id=2> <h3> Celebrate World Wish Day </h3> <p> Support
E. <h2 id=3> Help </h2>
F. <a id=4>  </a>
C.  Action: CLICK
``` |
| WikiHow (Linear) | ```
"Problem": "How to Add Captions to Tables in Microsoft Word",
"Solution Steps": [
    "Select the table to which you want to add a caption.",
    "Using your mouse, click and drag over the entire table to select it.",
    "Right-click (or ctrl-click) the table and select Insert Caption.",
    "Enter your caption.",
    "Type the caption for this table into the \"Caption\" field.",
    "Select a caption label.",
    "Customize your caption numbers (optional).",
    "Choose where to place your caption.",
    "Click the \"Position\" drop-down menu, and choose whether to place the
        caption above or below the table.",
    "Click OK to add your caption to the table.",
    "Format your captions."
]
``` |
| IcM (Dynamic) | ```
How to Investigate Service A-To-Service B Connection?

### Step 1: Check Pull Task Execution From the Cluster

The direct impact of connection failure is pull task execution will not
    work. If Service A can continue to pull from Service B, then the
    incident can be dismissed as false alarm, the feature owner can
    investigate further to see why Echo fails. This can be visualized by
    pull task count over time in the last 8 hours in the following query:
    ```kusto
    <Code Block>
    ```

Disregard the last data point, if the data point is always above zero, then
    consider the alert as false alarm. If the chart sometimes drops to
    zero one hour ago and the number is low in general (for instance less
    than 20), it means the customer traffic in the cluster is low. In this
    case, observe for a longer period of time. If the data point is zero
    consistently in the past 30 minutes, then it is a real problem, and
    please Check if Other Clusters In the Region are Impacted.  Otherwise,
    continue to observe since Service A is pulling Service B just fine.
``` |

Table 8: Examples of each dataset. For Mind2Web, although the test set has fixed options for each step, there are different execution methods for the same task on each website, so it is essentially dynamic.

| Trouble Shooting Guide: How to Investigate Service A-To-Service B Connection? | |
|---|---|
| Original | ### Step 0: Determine the Region and Cluster Name 
 The region and cluster name can be found in the incident title. 
 ### Step 1: Check Pull Task Execution From the Cluster 
 The direct impact of connection failure is pull task execution will not work. If Service A can continue to pull from Service B, then the incident can be dismissed as false alarm, the feature owner can investigate further to see why Echo fails. This can be visualized by pull task count over time in the last 8 hours in the following query: *** 
 Disregard the last data point, if the data point is always above zero, then consider the alert as false alarm. If the chart sometimes drops to zero one hour ago and the number is low in general (for instance less than 20), it means the customer traffic in the cluster is low. In this case, observe for a longer period of time. If the data point is zero consistently in the past 30 minutes, then it is a real problem, and please Check if Other Clusters In the Region are Impacted. Otherwise, continue to observe since Service A is pulling Service B just fine. 
 ... |
| Reformulated | ## 1.Check Pull Task Execution From the Cluster. 
 ### Prerequisite 
 The region and cluster name can be found in the incident title. 
 ### Header 
 Check Pull Task Execution From the Cluster 
 ### Body 
 Run the following query to check pull task execution from the cluster (please use the cluster name from the previous step) *** 
 ### Linker 
 - If the data point is always above zero, then consider the alert as false alarm.[MITIGATE] 
 - If the chart sometimes drops to zero one hour ago and the number is low in general, it means the customer traffic in the cluster is low. In this case, observe for a longer period of time.[MITIGATE] 
 - If the data point is zero consistently in the past 30 minutes, then it is a real problem, and please Check if Other Clusters In the Region are Impacted.[CONTINUE] 
 - Otherwise, continue to observe since Service A is pulling Service B just fine.[MITIGATE] 
 ... |
| Logic Unit | <pre>{
    "#type#": "step",
    "#meta data#": {
        "#title#": "How to Investigate Service A-To-Service B Connection",
        "#id#: "",
        "#date#": ""
    },
    "#prerequisite#": "The region and cluster name are given.",
    "#header#": "Check Pull Task Execution From the Cluster.",
    "#body#": "Run the following query to check pull task execution from
        the cluster (please use the cluster name from the previous step)
        :***",
    "#linker#": "If the data point is always above zero, then consider the
        alert as false alarm.[MITIGATE] If the chart sometimes drops to
        zero one hour ago and the number is low in general, it means the
        customer traffic in the cluster is low. In this case, observe for
        a longer period of time.[MITIGATE] If the data point is zero
        consistently in the past 30 minutes, then it is a real problem,
        and please Check if Other Clusters In the Region are Impacted.[
        CONTINUE] Otherwise, continue to observe since Service A is
        pulling Service B just fine.[MITIGATE]",
    "#default_parameters#": {
        "<TIME>": "",
        "<CLUSTER NAME>": ""
    }
}

...</pre> |

Table 9: An example of reformulated TSG and its corresponding Logic Unit of THREAD.

- Appendix: the supplement of the incident that is not important or labeled by TSG, usually providing additional resources, data, links and so on.

1. You need to identify each second-level subsection, including third-level subsection if needed, analyze its content or purpose, and categorize it accordingly. For those belonging to Step, you should capture the inner connections, such as Causality or Temporal relations, and present them in the correct order.
2. Your returned formulated TSG should be in JSON format. Make sure that the keys originate from these categories: Terminology, FAQ, STEP ad Appendix. Each value should be a list of dictionaries. The keys for them are "prerequisite", "header", "body", and "linker". All values within the lists need to align with the original context, with truthful meaning and necessary **code block**.
3. Importantly, the "linker" is used to imply the dual role of providing the action's result and connecting to the next step using the "if-then" sentence format. You should formulate each step's linker to be "If any results are obtained by executing the corresponding action in the previous step, then **the true intent of the following step** provided here". Implicit linkers like "proceed to the next step." or "then the intent of the following step should be taken into consideration." should be avoided.
4. For each "if" condition at every step in the STEP, it is necessary to add a special token behind the "then" condition within the "linker". The options for these tokens are "[CONTINUE]", "[CROSS]", and "[MITIGATE]". - The token "[CONTINUE]" indicates that the actions corresponding to this "if" condition are part of the continuum within the same TSG's STEPs. - The token "[CROSS]" signifies that the subsequent actions require a transition to a different set of steps that are external to the current TSG's STEPs. - The token "[MITIGATE]" implies that the actions following the "if" condition convey that the incident is mitigated, or necessitate communication with on-call engineers or teams.
The use of this special token is instrumental in verifying the completeness and structural integrity of the STEP section.

<TWO EXAMPLES HERE>

[User]
Here is the <TSG> you need to formulate:
{TSG}

---

**Instruction that extracts code template and default parameters from the source code.**

[System]
You are a helpful assistant that extracts the code template and the default parameters from the provided code instance in .  is a code block that contains several parameters. You should replace those parameters with placeholders and output the code template with placeholders and default parameters.

<ONE EXAMPLE HERE>

Your response should be in the JSON format as below:

```
{
    "#CODE_TEMPLATE#": where you replace the parameters in  with placeholders,
    "#DEFAULT_PARAMETERS#": where you keep the parameters in  as default values.
}
```

[User]
Here is the  you need to extract:
{CODE}

> **Instruction that selects the most relevant logic unit based on user query and chat history.**
>
> [System]
> You are a helpful assistant that selects the most relevant element from <LU_LIST> based on the user's query in <QUERY> and chat history in <CHAT_HISTORY>. Please respond with the JSON format.
> Each element in <LU_LIST> is in json format and contains the following fields:
>
> ```
> {
>     "#type#": "the type of the element, select from the following types: Terminology, FAQ,
>         Step, and Appendix.",
>     "#meta data#": "the description of the troubleshooting guide.",
>     "#prerequisite#": "The prerequisite of this step, before taking the current step, the
>         prerequisites should be finished.",
>     "#header#": "The information describes the intent of the <INFO>.",
>     "#body#": "The action is the content which troubleshoots the incident or explain the #
>         header#. the action may contain code blocks in markdown format, and parameters
>         are replaced with placeholders",
>     "#linker#": "the expected output after taking the #action#. It is defined in the
>         following format in markdown: -If **condition**, then **should_do**. It can
>         contain multiple if-then cases.",
>     "#default_parameters#": "the default parameters that could fill in placeholders in
>         code blocks in #body#."
> }
> ```
>
> - The elements in <LU_LIST> contain possible information that can answer the user's query in <QUERY>. However, they may not be all relevant to the query or useful to answer the user's query. You should select the most relevant element from the <LU_LIST> based on the user's query in <QUERY>.
> - In particular, you should focus on the following fields in the element: #header#, #body#. Most importantly, the <QUERY> needs to match with the #intent# and the #body# has to provide actions to reach the goal of the <QUERY>, please ignore the #linker# and do not map the <QUERY> with #linker#.
> - As you choosing from <LU_LIST>, you need to check if all the #prerequisite# are met in previous history. If the #prerequisite# is not finished, then it should not be chosen.
> - Try to select only one element from <LU_LIST>. If it is not possible to select only one element, you can select multiple elements from <LU_LIST>:
>
> ```
> [
>     {
>         "INDEX": the index of the element in <LU_LIST>.
>         "INTENT": the #header# of the element, the index starts from 0.
>         "EXPLANATION": justify why you select this node.
>     }
> ]
> ```
>
> - If there is no element in <LU_LIST> that can answer the user's query in <QUERY>, you should try to select the most relevant element to the user's query considering that the user might use the wrong terminology:
>
> ```
> [
>     {
>         "INDEX": the index of the element in <LU_LIST>.
>         "INTENT": the #header# of the element, the index starts from 0.
>         "REPHRASED_QUERY": the rephrased query that you think the user is asking about.
>         "EXPLANATION": justify why you select this node.
>     }
> ]
> ```
>
> - Unless you are confident that there is no element in <LU_LIST> that is even close to the user's query:
>
> ```
> {
>     "NO_INFO_EXPLANATION": where you give your explanation.
> }
> ```
>
> - Your answer should be in the JSON format in a list after <RESPONSE>.
>
> [User]
> <LU_LIST>: {LU_LIST}
> <QUERY>: {QUERY}
> <CHAT_HISTORY>: {CHAT_HISTORY}

## C.2 MIND2WEB

We show the details about the document generation instruction we use, the different formats of documents, and the examples we generate.

---

**Instruction that generates specific format of document for Mind2web dataset.**

[System]
You are adept at performing website navigation tasks, and you will be provided with simulation data from Mind2Web, designed for developing and evaluating generalist agents capable of following language instructions to complete complex tasks on any website.

The data includes a step-by-step execution process, each step encompassing HTML code, Tasks, Previous Actions, and the Element and Action of this step. Note that the Element comes from the HTML code, and if the correct action is not present on the current page, the Element is None, and you should retrieve it from next step's Previous Actions.

Now your task is to write a comprehensive and adaptable reference document that outlines the general process for completing tasks like the given task. This document should serve as a guide for others to perform similar tasks on the same website in the future. So it should not be limited but can use this data to be the example, and should be general enough.

Please return the complete reference document that adheres to these guidelines.

[User]
The format of the documents should be as follows: {FORMAT}
The given execution process is as follows: {EXAMPLE}

---

We follow Wang et al. (2024) to brainstorm diverse formats of documents for Mind2web dataset used for retrieval, and the results are listed in Table 10.

| Format | Description |
|---|---|
| Structured Markdown | - The document must be structured into sections in markdown format.
- It should include a task overview, introduction, process steps, and conclusion.
- Each step in the process includes detailed explanations for Intent, Prerequisite, HTML Code Reference, Action, Reason, and Result.
- The Prerequisite is to specify any conditions or prior actions that must be met or completed before proceeding with the current step in the process.
- Ensure that each step is explicitly connected to the next one, and the result is written in the "if-then" schema where the "Intent" of this step is completed, and the outcome "then" is the next step's Intent.
- The HTML Code Reference gives hints of the Action like some '<button>', '', or other elements or attributes. You need to use the given task as an example.
- The Action comes from "Click", "Type", "Hover", "Press Enter". |
| Hierarchical Guideline | - A structured text document with numbered steps for each task.
- Each step includes a title, description, the HTML code involved, and the action to be taken.
- Previous actions are referenced where necessary, with hyperlinks to the relevant steps.
- Appendices for HTML code references, glossary of terms, and FAQs. |
| Tabular Checklist | - A printable checklist with each task and subtask, including checkboxes for completion.
- Each checklist item includes a code snippet and the action required.
- A troubleshooting section that lists common problems and their solutions.
- Tips for what to do when the expected element or action is not available.
- References to more detailed instructions or external resources for complex tasks. |
| Narrative Document | An entire description of the execution process without special structures. |

Table 10: The description of different formats of documents on Mind2web.

For LU merge, we first identify the similar logic units by using the SpaCy library to calculate the textual similarity of LU headers. Then we leverage LLM to merge LUs with the following prompt:

---

**Instruction that merges logic units with similar header.**

[System]
You are tasked with a set of Logic Units that contain information about different steps in web navigation task. Each unit includes components like type, title, header, prerequisite, body, and linker. Some units have similar intents and can be merged to streamline the process and reduce redundancy.

Your task is to merge logic units with similar header into a single unit that combines their prerequisite, body and linker in a logical and coherent manner.

- Most importantly, as merging, you should concentrate on the linker, you need to unite the linker with similar intent, and carefully compare their "if" conditions. These conditions should now depend on the title specifics, guiding the user to the appropriate next action based on the context of the task.
- And for prerequisite, you should synthesize the prerequisites from the individual units, preserving the original logic and ensuring that the merged unit sets the necessary conditions for the subsequent steps.

The purpose of this merge is to create a more efficient set of instructions that can handle multiple scenarios without repeating steps.

Here is an example:

Please only return the merged unit in JSON format, keeping the same structure with the input.

[User]
The logic units you need to merge are as follows: {units}

---

## C.3 WIKIHOW

---

**Instruction that evaluates the generated answer compared with ground truth for Wikihow.**

[System]
You are a helpful and precise assistant for checking the quality of the answer. We would like to invite you to evaluate the performance of the system in answering a user's question in <Question>.

I will give you the answer generated by the system in <Generation> and the ground truth answer in <Ground Truth> respectively. Your evaluation will contain five sub-evaluation tasks:

1. Both two answers contain a list of steps. Your task is to extract action items from the provided steps in both answers. The action item is defined as a combination of action and element. Compare the action items to identify similarities. Output the similar action items. Count the count of similar action items.

- Your answer should contain the extracted two action item sets (in the format as a list of strings).
- Your answer should contain a set of similar action items (in the format of a list of strings). Similar action items are those sharing similar intent or achieving similar goals. Each similar action pair in the list should be in the format of "similar action item from action item set1 / similar action item from action item set2" - Your answer should contain the count of similar action items.

2. Can <Generation> completely solve the user's question?
- Your answer should be "Yes" or "No".
- Your answer should contain the reason(s) for your choice. You should not focus on the length of the answer or the details of the answer, but you should focus on whether the steps could solve

---

the user's question and the quality of the steps compared with the ground truth.

Your output should be in the following format in JSON:

```
{
    "Subtask1": {
        "Action items in Generation": ["action item 1", "action item 2", ...],
        "Action items in Ground Truth: ["action item 1", "action item 2", ...],
        "Similar action items": ["similar action item 1", "similar action item 2", ...],
        "Count of similar action items": 2
    },
    "Subtask2": {
        "Choice": "Yes" or "No",
        "Reason": "reason for your choice"
    }
}
```

[User]
Here is the user's question <Question>: {Question}
The answer from system <Generation> is: {Generation}
The ground truth answer <Ground Truth> is: {Ground Truth}

## D  DETAILS ABOUT RAG SYSTEM

We take the scenario of Mind2web to show the instructions we use in our RAG-based QA system.

---

**Instruction that is used for the baselines of RAG system on Mind2web.**

[System]
You are a helpful assistant who is great at website design, navigation, and executing tasks for the user. Now please proceed with the <CURRENT_STEP> and make your choice, remember that only based on the helpful document information from <DOC_CONTEXT> and the previous step chat history between user and assistant in <CHAT_HISTORY>.

Your response should be in the format of "Answer: C. Action: SELECT Value: Pickup".
The answer is A, B, C..., the Action comes from [CLICK, TYPE, SELECT] and the Value is not always needed.

[User]
<DOC_CONTEXT>: {DOC_CONTEXT}
<CHAT_HISTORY>: {CHAT_HISTORY}
<CURRENT_STEP>: {CURRENT_STEP}

---

**Instruction that is used for the RAG system utilizing THREAD Paradigm on Mind2Web.**

[System]
You are a helpful assistant who is great at website design, navigation, and executing tasks for the user. Now please proceed with the <CURRENT_STEP> and make your choice, remember that only based on the helpful structured document information from <LOGIC_UNIT>, and the previous step chat history between user and assistant in <CHAT_HISTORY>.

Your response should be in the format of JSON:

```
{
    "CHOICE": the choice you make from A B C ...,
    "ACTION": the corresponding action choosing from ['CLICK', 'TYPE', 'SELECT'],
    "VALUE": the corresponding value if needed,
    "INTENT": the intent of the next step, which should be retrieved and judged from the "
        if" conditions in #output# from <LU> according to the current step and actions
        and choose the corresponding "then" outcome, do not guess it based on current
        Task in <CURRENT_STEP> by yourself unless the <LU> is irrelevant to <CURRENT_STEP
        >,
}
```

[User]

```
<LOGIC_UNIT>: {LOGIC_UNIT}
<CHAT_HISTORY>: {CHAT_HISTORY}
<CURRENT_STEP>: {CURRENT_STEP}
```

# E  ADDITIONAL EXPERIMENTAL RESULTS

## E.1  COST AND SCALABILITY

We analyze the preprocessing overhead for constructing the final knowledge base using the Mind2Web dataset and compare THREAD to other paradigms that also leverage LLMs during preprocessing. From Table 11, THREAD achieves the highest element accuracy (68.29%) with a balanced trade-off between cost and performance. While its cost is higher than the Chunk and Semantic paradigms, it is significantly lower than the Proposition paradigm, making it suitable for real-world applications. Additionally, in Figure 4, we show that Llama3-70B achieves comparable performance with GPT-4, further emphasizing THREAD 's effectiveness and generality.

| Paradigm | Chat Tokens (I/O) / Doc | Embedding Tokens / Doc | Price / Doc ($) | Chat Model API Call / Doc | Ele. Acc |
|---|---|---|---|---|---|
| Recursive (Chunk) | - | 838 | 0.000084 | - | 64.23 |
| Semantic | - | 3802 | 0.00038 | - | 65.14 |
| Proposition | 61836 / 388 | 978 | 0.63 | 28.8 | 62.37 |
| THREAD | 2553 / 654 | 654 | 0.045 | 1 | 68.29 |

Table 11: Cost of different paradigms on Mind2web.

More importantly, to evaluate THREAD 's scalability, we curate datasets by transforming existing datasets (e.g., Mind2Web) and collecting data from the internet (e.g., WikiHow) and examined its performance on three datasets with different sizes. Table 12 shows that THREAD consistently outperforms the Chunk paradigm across datasets of varying scales, achieving higher element accuracy while maintaining acceptable costs. This demonstrates THREAD 's scalability in handling both longer documents and larger corpora.

| Dataset | Total Tokens | Original Tokens / Doc | Chat Tokens (I/O) / Doc | Embedding Tokens / Doc | Price / Doc ($) | Δ Ele. Acc |
|---|---|---|---|---|---|---|
| Incident Mitigation | 112K | 2002 | 2814 / 1464 | 1464 | 0.072 | 23.68%-36.68% ↑ |
| WikiHow | 200K | 2066 | 4608 / 778 | 778 | 0.069 | 44.3%-51.54% ↑ |
| Mind2Web | 410K | 838 | 2553 / 654 | 654 | 0.045 | 4.06% ↑ |

Table 12: Scalability across different scenarios.

## E.2  GENERALIZATION TO OTHER 5WS QUESTIONS

To evaluate whether THREAD can handle '5Ws' questions without compromising performance, we conduct experiments on 2WikiMultiHopQA [14]. Specifically, we sample 200 questions of two types, 'inference' and 'compositional' type, like "Who is the maternal grandfather of Abraham Lincoln?" and "Who is the founder of the company that distributed La La Land film?" In our experiment, we use the relations The findings from Table 13 demonstrate that THRED not only surpass the chunk-based paradigm on multi-hop questions, but also excels at handling more complex how-to question.

| Method | F1 | EM | Acc |
|---|---|---|---|
| Chunk | 24.23 | 15.50 | 25.00 |
| THEAD | 44.77 | 30.00 | 46.00 |

Table 13: Experimental results on the generalization of THREAD to 2WikiMultiHopQA.

---

[14] https://github.com/Alab-NII/2wikimultihop

## F  LIMITATIONS

This work focuses on evaluating the effectiveness of THREAD by designing how-to questions in three specific scenarios, covering both linear and dynamic how-to questions. However, there are several limitations that point to future directions. First, as our logic-based knowledge base can coexist with the original chunk-based knowledge base, we do not extend our method to factoid questions, such as multi-hop or long-form questions. Second, the experiments primarily use ChatGPT and LLaMA as the backbones. Future work should include evaluations with other LLMs and alternative retrievers such as Contriever to further validate the effectiveness of THREAD. Lastly, while extracting logic units involves an initial cost in terms of LLM usage, this is a one-time process. Once the knowledge base is constructed, it provides significant advantages for industrial applications, particularly in terms of subsequent updates and maintenance.

