# OpenReview forum: "Thread: A Logic-Based Data Organization Paradigm for How-To Question Answering with Retrieval Augmented Generation"
_ICLR.cc/2025/Conference — Submitted to ICLR 2025_

### Official Review · Reviewer_pBXn · 2024-10-28

**Soundness:** 4
**Presentation:** 3
**Contribution:** 3
**Rating:** 8
**Confidence:** 3

**Summary:**

This paper proposes a novel data organization paradigm named THREAD, aimed at improving the capability of question answering systems in dealing with "how-to" questions, particularly those requiring dynamic, step-by-step solutions. Existing retrieval-augmented generation systems face challenges in connecting documents or chunks when handling such questions. THREAD introduces a new granularity of knowledge known as "logical unit" and adaptively utilizes the Linker to explicitly represent the internal logic between texts. This enables the method to exclude redundant information during the reasoning process and better maintain the coherence of answers. The paper conducts extensive experiments in open-domain and industrial settings, including Web Navigation, Wikipedia Instructions, and Incident Mitigation scenarios. Additionally, the authors provide a detailed analysis of different data organization paradigms, including ICL, SL, RAG based on various chunking approaches and the proposition, etc, to evaluate the superiority of the THREAD paradigm compared to existing methods. This paper addresses a specific and practical problem, proposes a well-motivated approach, and supports its claims through rigorous experimentation.

**Strengths:**

Originality:
This paper introduces THREAD, an innovative data organization paradigm that captures the logical structure within documents and the connections between steps through the concepts of logical units and linkers.

Quality:
In terms of experimental design, the paper conducts extensive testing on datasets from both open-domain and industrial environments, with detailed comparisons to existing methods, demonstrating the generalization ability and reliability of its approach. The paper also considers language models of different sizes.

Clarity:
The paper is well-structured, from the problem statement to the methodology, experimental design, and results analysis. The presentation of experimental results is clear, and the use of charts and figures makes the data easy to compare and understand.

Importance:
This paper effectively addresses "how-to" problems in practical applications, demonstrating advantages compared to existing methods.

**Weaknesses:**

1. The description of the extraction and merging process of logical units in the paper is somewhat vague, lacking an in-depth explanation: Although Sections 3.1 and 3.3 of the paper provide introductions, they mainly focus on the functions and roles, with relatively vague descriptions of specific implementation strategies, especially for the crucial Linker component. Additionally, the specific implementation process of unit merging is not clear, such as how to merge the various parts of two logical units.
2. The THREAD proposed in the paper represents a novel paradigm specifically tailored for addressing "how-to" questions, yet it poses significant implementation challenges in other real-world scenarios, particularly in establishing Linkers between logical units, and exhibits limited method scalability. The paper primarily focuses on how to better extract logical units, while the precise recognition and extraction of relationships between any two text segments or documents are not discussed in detail.

**Questions:**

1. Is the extraction of the Linker component based on a single document or a comprehensive analysis of multiple documents?
2. Can the paper provide a clearer explanation of the unit merging process, including how four parts of two logical units are merged?
3. When performing logical unit extraction, how is content extracted from a document that may contain multiple aspects and themes, especially when the specific question is unknown?

---

> ### Author Response · Authors · 2024-11-22
> **Response to Reviewer pBXn [1/2]**
>
> > ### W1: The description of the extraction process of logical units in the paper is somewhat vague, lacking an in-depth explanation:  ...especially for the crucial Linker component...
>
> Thank you for highlighting this point. We would like to provide additional clarification regarding the extraction process.
>
> This process involves a two-stage approach. Firstly, documents are reformulated into structured formats. Then, logic units are extracted with explicit definitions for each component, particularly the crucial Linker. This component serves a dual role: linking conditions across steps and ensuring structural integrity.
>
> To ensure consistency, we use a special token after each condition in the Linker, validating the completeness and alignment of the extracted STEP LUs. Additionally, the format of the Linker is explicitly constrained to maintain uniformity.
>
> For further details, we refer to Section 3.3 and Appendix 3.1, where we provide the reformulation prompts used in our experiments. Specifically, definitions of LU components and types are detailed in Lines 745–748 and 864–875, while constraints and formatting of the Linker are outlined in Lines 876–880. The use of the special token is explained in Lines 881–888.
>
> We hope this explanation clarifies the implementation and addresses your concerns effectively.
>
> > ### W2(a): The THREAD proposed in the paper represents a novel paradigm specifically tailored for addressing "how-to" questions, yet it poses significant implementation challenges in other real-world scenarios, particularly in establishing Linkers between logical units, and exhibits limited method scalability.
>
> > ### W2(b): The paper primarily focuses on how to better extract logical units, while the precise recognition and extraction of relationships between any two text segments or documents are not discussed in detail.
>
> Thanks for your thoughtful questions. We would like to address your concerns regarding the recognition and extraction of relationships between text segments or documents, as well as the scalability and implementation challenges of THREAD in real-world scenarios.
>
> - As answering in W1, we rely on LLMs to reformulate and extract logic units from documents. The coherence and internal logic of the documents are derived through the parsing and context-understanding capabilities of LLMs, as research [1] shows that LLMs excel in parsing structured information from unstructured data.
>
> - Manually reformulating documents to extract logic units is labor-intensive and costly. In-context learning offers a more efficient and scalable alternative, enabling LLMs to perform this task effectively without extensive manual intervention.
>
> - However, establishing linkers between logic units largely depends on the quality of the source documents.
>
>   - Our work focuses on extracting existing internal logic rather than generating new relationships that do not exist in the source documents.
>   - Creating relations between LUs that are not present in the documents is beyond the scope of this work and is not the focus of THREAD.
> - Lastly, we have validated THREAD’s effectiveness through experiments across three diverse scenarios, demonstrating its capability to handle "how-to" questions. Our design is generalizable and can be easily extended to other scenarios with minimal adjustments.
>
> We hope this response clarifies your concerns. If you have further questions, we would be happy to discuss them in more detail.
>
> [1] Xu D, Chen W, Peng W, et al. Large language models for generative information extraction: A survey[J]. Frontiers of Computer Science, 2024, 18(6): 186357.

---

> ### Author Response · Authors · 2024-11-22
> **Response to Reviewer pBXn [2/2]**
>
> > ### Q1: Is the extraction of the Linker component based on a single document or a comprehensive analysis of multiple documents?
>
> Thank you for your question. The extraction and processing of the Linker component involve both single-document analysis and multi-document integration:
>
> - **Initial extraction based on a single document**
>
>   The extraction of LU begins with analyzing individual documents. For each document, we extract LUs based on the document's internal logical flow. Specifically, in this process, some documents may reference other documents, and the Linker in certain LUs might directly establish cross-document relationships.
>
> - **LU Merge involves multi-document integration**
>
>   In LU merge, LUs from different documents are compared, LUs with identical or similar header are unified to form a single LU, as shown in Figure 2(b). At this point, the Linkers associated with these LUs will be integrated. This step is crucial when multiple documents address the same question or follow similar steps, ensuring cohesive relationships across documents.
>
> More details on the LU merging process can be found in the following responses. We hope this explanation clarifies the point.
>
>
> > ### W1 & Q2: the specific implementation process of unit merging is not clear, such as how to merge the various parts of two logical units.
>
> Here is a detailed explanation of LU Merge process, and we have provided this information in revised version (refer to Appendix 3.2):
>
> **1. Identifying Similar Logic units**
>
> As written in Lines 256-257, we use the SpaCy library to calculate the textual similarity of LU headers. LUs with similar headers are grouped as candidates for merging.
>
> **2. Merging Logic units with LLMs**
>
>   We leverage LLMs to merge them, guided by the following criteria:
>
>   - Linker Coherence: Focus on uniting the linkers with similar intents while carefully comparing their "if" conditions. These conditions are adjusted to depend on the specifics of the original questions being addressed, ensuring the resulting LU guides users to the appropriate next action based on the context of question.
>
>   - Prerequisite Synthesis: Combine the prerequisites from individual LUs while preserving their original logic. The merged LU sets the necessary conditions for subsequent steps.
>
>   - Body Integration: The actions in the LU body are compared and integrated, ensuring they are consistent and aligned with the merged linkers and prerequisites.
>
> By unifying LUs with similar headers, this process minimizes duplication and enhances the coherence of the overall problem-solving framework.
>
> > ### Q3: When performing logical unit extraction, how is content extracted from a document that may contain multiple aspects and themes, especially when the specific question is unknown?
>
> Thank you for the question. Below, we clarify how logic units (LUs) are extracted from documents containing multiple aspects and themes, even when the specific question is unknown:
>
> - **Documents with multiple aspects and themes**
>
>   The presence of multiple aspects and themes does not hinder the reformulation process. For instance, dynamic how-to questions often involve documents with tree-structure solutions, where each path represents a distinct solution. Our paradigm extracts each node (step) as a separate logic unit, ensuring comprehensive coverage of the document’s content.
> - **Unknown specific question**
>
>   The extraction process itself remains unaffected by the absence of a specific question, as it focuses on procedural details rather than question relevance. However, this can complicate the retrieval stage. In our experiments, we typically concatenate the question with the document header to improve retrieval. Without a known question, matching the extracted LUs to the correct query becomes more challenging, leading to reduced recall. However, the solutions described in the document often allow us to infer the intended question. This inference can help mitigate the impact on retrieval performance.
> We hope this explanation addresses your concerns effectively.

---

> ### Author Response · Authors · 2024-11-26
> **Kindly request your reply**
>
> Dear Reviewer pBXn,
>
> Thank you for your valuable feedback and constructive comments. We kindly ask if you could let us know whether our responses address your concerns and adjust your overall rating accordingly.
>
> If there are any remaining issues or clarifications needed, we would be happy to address them promptly.
>
> Sincerely,
>
> Authors of Submission 5754

---

### Official Review · Reviewer_oQG8 · 2024-11-02

**Soundness:** 2
**Presentation:** 3
**Contribution:** 2
**Rating:** 5
**Confidence:** 4

**Summary:**

The paper proposes THREAD, a novel data organization paradigm aimed at enabling current systems to handle how-to questions more effectively. Specifically, we introduce a new knowledge granularity, termed ‘logic unit’, where documents are transformed into more structured and loosely interconnected logic units with large language models. Extensive experiments are conducted across both open-domain and industrial settings.

**Strengths:**

The paper is written well and points out a good question.
The idea of LU is reasonable.

**Weaknesses:**

Objectively speaking, most RAG objects are chunks. This paper does not change this basic background. So the beginning of this article is very misleading. It is recommended not to exaggerate the motivation in the respect of chunks.

The last sentence of contribution # 3 does not seem to be supported by any experiments.

**Questions:**

1. The proposed method calls many logic units which act like tools and divide a question to many sub questions or new questions. So The proposed method should be compared with baselines using tool learning, such as Toolllm and ControlLLM, etc. You can find more from "Tool Learning with Large Language Models: A Survey".

2. How to ensure that there are tight connections between LUs? It is easy to form a large number of meaningless nodes. If one LU triggers more than one other LUs, or if a problem triggers more than one LU, how does your approach work? What if none of the Linkers support the answer to the question?

3. The idea of LU is good, but the construction of LU in practical applications is very troublesome. LU is completely based on human design and is very complex. Do you have any ideas to deal with it? What is the contribution to the performance of each part in LU? The meta data in LU did not described in the Fig.2.

4. The baselines in this paper are not particularly persuasive(some are not based on RAG), and the outcome of LU should be tested in more related baselines.

5. The data sets used in this article are all small-scale data sets. Have you tried large-scale data sets?

---

> ### Author Response · Authors · 2024-11-23
> **Response to Reviewer oQG8 [1/3]**
>
> ### **Weakness**
> > ### W1: Objectively speaking, most RAG objects are chunks. This paper does not change this basic background. So the beginning of this article is very misleading.
>
> Thank you for your feedback. However, we find the claim that "our paper is misleading or exaggerates the motivation regarding chunks" unclear and would appreciate further clarification.
>
> First, our introduction begins by **analyzing the limitations of current RAG methods, which predominantly focus on addressing 5W questions**, while largely overlooking how-to (1H) questions. We then highlight the prevalent chunk-based paradigm fails to effectively address how-to questions and propose a new paradigm, THERAD, as a solution.
>
> Second, **the diversity in retrieval granularity among existing RAG methods supports our motivation**. According to a recent survey [1], Table 1 provides an overview of these methods, and the "Retrieval Granularity" column shows that while around 50% of methods are chunk-based, others rely on alternative granularities, such as sentences, documents, or triples (as referenced in Subscript 2 of our paper). This diversity in retrieval granularity highlights the proposal of our logic unit.
>
> Given this context, we do not see how the paper "does not change this basic background" or "exaggerates the motivation" regarding chunks. Instead, we believe our work fairly critiques chunk-based paradigms and introduces a novel approach to handling how-to questions.
>
> We would appreciate further clarification if there are any aspects that are misleading.
>
> [1] Gao Y, Xiong Y, Gao X, et al. Retrieval-augmented generation for large language models: A survey[J]. arXiv preprint arXiv:2312.10997, 2023.
>
> > ### W2: The last sentence of contribution # 3 does not seem to be supported by any experiments.
>
> Our contribution #3 encompasses two key aspects:
>
> - **THREAD significantly outperforms existing data organization paradigms across three scenarios.**
>
>   From Section 5.1, we provide our main results across three different scenarios, our Thread consistently compasses the chunk-based or doc-based methods. More results can be found in Table 2, 3, 4.
>
> - **THREAD requires less information, while effectively handling various document formats and substantially reducing the knowledge base size.**
>
>   - As detailed in Sections 5.3 and 5.4, we compare THREAD against existing paradigms, including chunk, document, semantic chunk, and proposition. The analysis shows that THREAD achieves the highest performance (over 5% higher Element Accuracy) while requiring the fewest tokens—only one-fourth of the tokens needed for chunk-based methods, as demonstrated in Table 6.
>
>   - Additionally, THREAD demonstrates robustness in handling diverse document formats. In Table 7, THREAD consistently outperforms chunk-based paradigms across different document formats.
>
>   - Finally, Table 1 compares the number of chunks and LUs in the knowledge base, validating that THREAD significantly reduces the size of the knowledge base.
>
> We hope this explanation clarifies THREAD’s contribution.

---

> ### Author Response · Authors · 2024-11-23
> **Response to Reviewer oQG8 [2/3]**
>
> ### **Question**
> > ### Q1: The proposed method calls many logic units which act like tools and divide a question to many sub questions or new questions. So The proposed method should be compared with baselines using tool learning, such as Toolllm and ControlLLM, etc.
>
> Thank you for raising this question. We would like to clarify some misunderstandings regarding the nature of our proposed method and its relation to tool learning.
>
> 1. **Tool learning and RAG address fundamentally different topics**
>
>    **Tool learning focuses on selecting and leveraging a limited set of predefined tools for specific tasks**. In contrast, THREAD is a new data organization paradigm used for RAG to better address how-to questions. The claim 'calls many logic units which act like tools' is a mischaracterization. At each step, the system retrieves only one LU, generates corresponding actions, and dynamically determines the next step based on the execution outcome. It does not work like calling tools during the process.
>
> 2. **THREAD is a data organization paradigm**
>
>    Since our proposal is a paradigm, our comparisons are appropriately conducted against existing paradigms like chunks, documents, and propositions. In Section 5, we perform comprehensive experiments across diverse scenarios to demonstrate THREAD’s advantages.
>
> 3. **Different usage of Tools**
>
>    Tool learning emphasizes **when and how to call appropriate tools to complete tasks**, such as mathematical reasoning or coding. **In contrast, our industrial setting** (incident mitigation) involves executing predefined tools or commands as specified in the documents. These **tool calls do not require learning or adaptation**. Instead, the core of THREAD lies in generating accurate, step-by-step instructions tailored to the query.
>
> We hope this explanation clarifies the distinctions between THREAD and tool learning. Thank you again for your insightful question.
>
> > ### Q2: How to ensure that there are tight connections between LUs? It is easy to form a large number of meaningless nodes. If one LU triggers more than one other LUs, or if a problem triggers more than one LU, how does your approach work? What if none of the Linkers support the answer to the question?
>
> Thank you for your thoughtful question. Below is our response to clarify the concerns:
>
> - **The mechanism of triggering multiple LUs**
>
>    It is natural for a single problem to involve multiple LUs. Since we aim at solving how-to question step-by-step, as illustrated in Figure 3, the workflow requires only one LU for each step and dynamically determines the next step based on the results of the current one.
>
>    This process is managed by matching the conditions defined in the "linker" component of the current LU to the output of the current step. If none of the linkers match, it indicates that the problem cannot be resolved with the existing knowledge base. This situation highlights either a limitation in the knowledge base or the need to introduce additional LUs to fill the gap.
>
> - **Clarification on 'meaningless nodes'**
>
>    The connections between LUs are derived from the inherent logical structure of the documents themselves. To ensure meaningful organization, we have identified and implemented a diverse set of LU types based on real-world scenarios, including terminologies, FAQs, and Steps (in Section 3.2). These predefined LU types help prevent the creation of irrelevant or "meaningless" nodes.
>
> However, we are unclear about the specific definition of "meaningless nodes" in this context. Does it refer to LUs with empty linkers, or to nodes of a specific type?
>
> We would appreciate further clarification to address this concern more thoroughly.
>
> > ### Q4: The baselines in this paper are not particularly persuasive(some are not based on RAG), and the outcome of LU should be tested in more related baselines.
>
> Thank you for your question.
>
> In Section 4.3, we outline the baseline settings used in our experiments. To highlight the effectiveness of RAG methods in handling complex questions, we include baselines such as in-context learning on the Mind2web dataset, as mentioned in Line 417.
>
> However, the primary focus of this work is our proposed data organization paradigm, THREAD. The experiments compare
> THREAD against other paradigms across various scenarios to demonstrate its effectiveness.
>
> Thus, the critique that "the baselines are not particularly persuasive" does not hold. Similarly, the suggestion to test LU outcomes on additional baselines is unnecessary, as our experiments are designed to evaluate the core contribution: the performance of THREAD within the proposed paradigm.

---

> ### Author Response · Authors · 2024-11-23
> **Response to Reviewer oQG8 [3/3]**
>
> > ### Q3: The idea of LU is good. LU is completely based on human design and is very complex. Do you have any ideas to deal with it? What is the contribution to the performance of each part in LU? The meta data in LU did not described in the Fig.2.
>
> Thank you for recognizing the value of our proposed LU. Below are clarifications regarding its design, contribution, and metadata:
>
> 1. **Generability of LU Design and Construction**
>
>    The LU is designed to be straightforward yet comprehensive, aiming for generalizability and adaptability across various scenarios. While LU construction involves human design, its structure is intentionally reusable and flexible, minimizing the need for extensive reconfiguration when applied to new scenarios. This approach ensures that the complexity is managed effectively and serves a broader range of applications.
>
> 2. **Contribution of Each LU Component is shown in Section 4.3**
>
>    In Section 4.3, we provide a detailed explanation of how the LU components work together in RAG systems.
>
>    As shown in Figure 3, THREAD integrates LUs into RAG systems through the following workflow:
>
>    - Retrieval: Relevant LUs are retrieved based on query-header similarity.
>    - Filtering: Prerequisites filter out LUs that do not meet current conditions.
>    - Action Generation: The selected LUs are passed to LLMs, which use the Body component to generate actions.
>    - Linking: The Linker matches execution results to generate a new query for the next retrieval iteration.
>
>    These components collectively contribute to the system’s ability to generate step-by-step solutions for how-to questions.
>
> 3. **Clarity Regarding Meta Data**
>
>    The metadata in an LU is primarily used to facilitate updates when documents are modified. Since metadata does not directly impact the problem-solving process or how-to questions, it is not depicted in Figure 2. However, its role is important in ensuring efficient knowledge base updates, as described in Lines 258-260.
>
> We hope this addresses your concerns and provides clarity on the LU’s design and functionality.
>
> > ### Q5: The datasets used in this article are all small-scale data sets. Have you tried large-scale data sets?
>
> Thank you for raising this question.
>
> Currently, **there are no open-source datasets available for evaluating the performance on how-to questions**, making data collection a particularly challenging task.
>
> For this work, **we have curated our datasets by transforming existing datasets (e.g., Mind2Web) and collecting data from the internet (e.g., WikiHow)**, ensuring their relevance to the problem at hand. To support future work in this area, **we will release documentation for two publicly available datasets**, with the exception of IcM, which remains confidential due to its sensitive nature.
>
> The table below summarizes the results across three datasets of varying sizes:
>
> Dataset | Total Tokens Of Dataset  | Original Tokens / Doc | Chat tokens (I/O) / Doc  | Embedding tokens / Doc | Price / Doc ($) | Ele. Acc Improvement (THREAD v.s. Chunk)
> ---- | ---- | ---- | ---- | ---- | ---- | ---- |
> Incident Mitigation | 112K  | 2002 | 2814/1464 | 1464 | 0.072 | 23.68%～36.68% ↑
> WikiHow | 200K | 2066 | 4608/778 | 778 | 0.069 | 44.3%～51.54% ↑
> Mind2Web | 410K | 838 | 2553/654 | 654 | 0.045 | 4.06% ↑
>
> Our findings demonstrate that THREAD consistently outperforms the Chunk paradigm across datasets of various scales, achieving significantly higher element accuracy while maintaining reasonable costs. **This highlights THREAD's scalability and effectiveness in handling both longer documents and larger corpora**.
>
> If you are aware of any large-scale datasets suitable for evaluating how-to questions, we would greatly appreciate your recommendations and are eager to explore them.

---

> ### Author Response · Authors · 2024-11-26
> **Kindly request your reply**
>
> Dear Reviewer oQG8,
>
> Thank you for your valuable feedback and constructive comments. We kindly ask if you could let us know whether our responses address your concerns and adjust your overall rating accordingly.
>
> If there are any remaining issues or clarifications needed, we would be happy to address them promptly.
>
> Sincerely,
>
> Authors of Submission 5754

---

### Official Review · Reviewer_yhzo · 2024-11-03

**Soundness:** 2
**Presentation:** 3
**Contribution:** 3
**Rating:** 5
**Confidence:** 2

**Summary:**

This paper addresses the limitations of current RAG systems in answering "how-to" questions, which are often complex, multi-step, and dynamic in nature. The authors propose THREAD, a novel data organization paradigm that restructures documents into interconnected "logic units". THREAD enables a more coherent, stepwise approach to answering how-to questions by capturing the logical flow within documents. Experiments demonstrate that THREAD outperforms existing chunk and document-based paradigms.

**Strengths:**

1.	THREAD introduces a novel logic-based organization paradigm that emphasizes the logical flow needed to handle complex how-to questions effectively.
2.	The paper’s design of logic units with structured components is well-thought-out. Each component serves a specific purpose, enhancing logical continuity and allowing for more coherent, step-by-step responses.
3.	Experimental results show that THREAD outperforms traditional data organization paradigms.
4.	THREAD reduces the number of retrieval units and token length required for generation, optimizing memory usage and computational efficiency.

**Weaknesses:**

1. The experiments mainly compare THREAD with chunk-based and document-based paradigms, with limited discussion on other advanced retrieval techniques, such as graph-based retrieval or hierarchical indexing.
2. In the industrial setting, the evaluation relies on human engineers. However, the paper does not provide enough details about the evaluation protocols, inter-annotator agreement, or potential biases.
3. THREAD’s effectiveness appears to depend on the assumption that documents are reasonably structured and easy to parse into logic units. The paper does not discuss THREAD’s robustness when faced with inconsistent or poorly formatted documents, which are common in real-world scenarios.
4. Although THREAD reduces the knowledge base size and required information for generation, the paper lacks a detailed analysis of computational efficiency and scalability when handling very large knowledge bases, potentially limiting its practicality for large-scale applications.
5. The introduction of multiple components and types in each logic unit may increase the complexity of maintaining the knowledge base, especially when documents are updated. While the paper mentions LU updating, it lacks sufficient detail on how updates will be managed and the potential maintenance overhead.
6. The paper does not include a detailed error analysis to identify common failure cases. Understanding these would help clarify THREAD's limitations and guide future improvements.

**Questions:**

None

---

> ### Author Response · Authors · 2024-11-23
> **Response to Reviewer yhzo [1/3]**
>
> > ### W1: compare THREAD with other advanced retrieval techniques, such as graph-based retrieval or hierarchical indexing.
>
> Thank you for raising such concern about other advanced retrieval techniques. Here we are willing to address it.
> - **Comparison with other advaced retrieval techniques**
>
>   Our experiments focus on comparing THREAD with the most commonly used data organization paradigms, such as document-based and chunk-based. According to the survey [1], over 80% of RAG methods rely on documents or chunks. Additionally, we **evaluate THREAD against the latest data organization techniques in Table 6**, such as proposition[2] and semantic chunk[3]. THREAD consistently outperforms these advanced approaches, achieving improvements ranging from 3.15% to 5.92% in Ele. Acc and 2.01% to 5.16% in Step SR.
>
> - **Reasons for not comparing with graph-based retrieval and hierarchical indexing**
>
>   While graph-based and hierarchical indexing methods are important retrieval techniques, they are not the focus of our work due to their limitations in addressing how-to questions. These limitations are discussed in the subsection Data Organization Paradigm in RAG within our related work section:
>
>   - Graph-Based Retrieval: This approach typically requires constructing a symbolic knowledge graph (e.g., entity-to-entity relationships), as noted in Lines 117–119, which is effective for 5W questions but is **inherently unsuitable for capturing the procedural logic and dynamic nature for how-to questions**. Moreover, building and maintaining a knowledge graph is resource-intensive, making it impractical for our target scenarios.
>
>   - Hierarchical Indexing: Hierarchical indexing **still essentially segments documents into chunks and organizes them hierarchically**. As mentioned in Lines 116-118, chunk-based approach ignores the logical and relational connections between chunks, potentially disrupting the inherent logic flow in document, thus, not aligned with the specific challenges THREAD is designed to address.
>
> And we thank you for your valuable suggestion, and will highlight discussion of graph-based retrieval and hierarchical indexing in our related work section.
>
> [1] Gao Y, Xiong Y, Gao X, et al. Retrieval-augmented generation for large language models: A survey[J]. arXiv preprint arXiv:2312.10997, 2023.
>
> [2] Chen T, Wang H, Chen S, et al. Dense x retrieval: What retrieval granularity should we use?[J]. arXiv preprint arXiv:2312.06648, 2023.
>
> [3] Greg Kamradt. The 5 levels of text splitting for retrieval, 2024.
>
> > ### W2: industrial setting, the details about human evaluation. Like evaluation protocols, inter-annotator agreement or potential biases.
>
> As described in Lines 322–333 and further clarified in Subscript 5, we conduct human evaluations involving both new-hire and experienced OCEs. They have an internal evaluation standard for incident mitigation, focusing on two main criteria: (1) whether each step includes the necessary operations, such as correct code, and (2) whether the incident is successfully mitigated.
>
> Upon analyzing the collected user data, we observe an issue with one new-hire OCE who has three failed records out of five incidents due to an authentication problem. Specifically, the execution engine requires OCE authentication to execute LLM-generated code for querying data. However, as a new-hire OCE, he/she lacks access to certain tables, leading to action execution failures and missing data. To ensure the integrity of the evaluation, we exclude this OCE's contaminated data from the analysis for fairness.

---

> ### Author Response · Authors · 2024-11-23
> **Response to Reviewer yhzo [2/3]**
>
> > ### W3: The paper does not discuss THREAD’s robustness when faced with inconsistent or poorly formatted documents, which are common in real-world scenarios.
>
> Thank you for highlighting the concern regarding THREAD’s robustness when dealing with inconsistent or poorly formatted documents. We need to clarify this point and address potential misunderstandings.
>
> - Firstly, we **do not assume that documents are inherently well-structured or easy to parse into logic units**. On the contrary, we **recognize the variability in document formats** and propose the 'Reformulation' process to reformulate the documents into structured ones as shown in Figure 2 (a).
>
> - Secondly, we **have explicitly designed experiments to evaluate THREAD’s robustness and generalizability across diverse formats**.
>    - In Section 4.2, we collect documents for Web Navigation scenarios in various formats, including markdown files, guidelines, and narrative documents. **Table 7 showcases THREAD’s robustness across these diverse formats**. For instance, while the chunk-based paradigm experiences a significant drop in performance on narrative documents (achieving 56.63 in Ele. Acc), THREAD demonstrates much higher score with 66.24.
>    - Moreover, THREAD’s performance on narrative documents is only marginally lower than its performance on structured markdown documents (68.29 in Ele. Acc), which underscores THREAD's ability to handle inconsistent or poorly formatted documents effectively.
>
> Hope our explanation addresses your concerns.
>
> > ### W5: it lacks sufficient detail on how updates will be managed and the potential maintenance overhead.
>
> Thank you for raising this question. We would like to clarify that the mechanism for updating THREAD is designed to be both straightforward and efficient.
>
> As stated in Lines 258-260, updates are managed using metadata to directly locate and modify the relevant logic units. This targeted approach eliminates the need to update the entire knowledge base, which is comparable to that of chunk-based paradigms, as both rely on metadata for precise updates.
>
> Therefore, THREAD ensures that only the affected units are updated, and will not increase maintenance overhead compared with chunk-based paradigms.
>
> > ### W6: The paper does not include a detailed error analysis to identify common failure cases. Understanding these would help clarify THREAD's limitations and guide future improvements.
>
> Thank you for pointing this out. We have conducted an early-stage error analysis by examining a subset of error cases from the Mind2Web dataset and categorizing the primary sources of errors as follows:
>
> - Missing LUs (15%): No appropriate LU is available.
> - Linker Issues (10%): The selected LU does not include the correct branch for the current step.
> - Retrieval Errors (20%): Failure to retrieve the correct LU, particularly in tasks with multiple valid paths.
> - LU Selection Errors (35%): Incorrect selection of the next step due to relying solely on the Top-1 LU based on embedding similarity.
> - Other Errors (20%): None of the above, such as LLM making mistakes.
>
> While errors related to missing LUs and linker issues largely depend on the quality of the source documents, we addressed the predominant "LU Selection Errors" by **introducing a prerequisite component within each logic unit and implementing a selector (LLM) to determine the most suitable LU**.
>
> By ensuring that prerequisite conditions are satisfied, rather than relying solely on similarity metrics, this approach effectively filters out irrelevant or unsatisfied units, significantly enhancing selection accuracy.
> The impact of these improvements is demonstrated in the ablation studies presented in Table 5.
>
> And we appreciate your valuable suggestion and will incorporate a more detailed error analysis in future iterations to further enhance THREAD's performance.

---

> ### Author Response · Authors · 2024-11-23
> **Response to Reviewer yhzo [3/3]**
>
> > ### W4: the paper lacks a detailed analysis of computational efficiency and scalability.
>
> Thank you for your thoughtful question regarding the scalability and cost-effectiveness of THREAD.
>
> - **Cost and performance analysis**
>
> We analyze the preprocessing overhead for constructing the final knowledge base using the Mind2Web dataset and compare THREAD to other paradigms that also leverage LLMs during preprocessing. The results are summarized as follows:
>
> Paradigm | Chat tokens (I/O) / Doc  | Embedding tokens / Doc | Price / Doc ($) |Chat Model API Call times / Doc | Ele. Acc
> -------- | -----  | ----- | ----- | ----- | -----
> Recursive (Chunk) | / | 838 |0.000084 | / | 64.23
> Semantic | / | 3802 | 0.00038 | / | 65.14
> Proposition | 61836/388 | 978 | 0.63 | 28.8 | 62.37
> THREAD | 2553/654 | 654 | 0.045 | 1 | 68.29
>
> From this table, THREAD **achieves the highest element accuracy (68.29%) with a balanced trade-off between cost and performance**. While its cost is higher than the Chunk and Semantic paradigms, it is significantly lower than the Proposition paradigm, making it suitable for real-world applications. Additionally, in Section 5.4, we show that Llama3-70B achieves comparable performance with GPT-4, further emphasizing THREAD’s effectiveness and generability.
>
> - **Scalability to Larger Datasets**
>
> More importantly, there are currently no open-source datasets available for evaluating the performance on how-to questions, making data collection a particularly challenging task.
>
> To evaluate THREAD’s scalability, we have curated datasets by transforming existing datasets (e.g., Mind2Web) and collecting data from the internet (e.g., WikiHow) and examined its performance on three datasets with different sizes. The table below summarizes the results:
>
> Dataset | Total Tokens Of Dataset  | Original Tokens / Doc | Chat tokens (I/O) / Doc  | Embedding tokens / Doc | Price / Doc ($) | Ele. Acc Improvement (THREAD v.s. Chunk)
> ---- | ---- | ---- | ---- | ---- | ---- | ---- |
> Incident Mitigation | 112K  | 2002 | 2814/1464 | 1464 | 0.072 | 23.68%～36.68% ↑
> WikiHow | 200K | 2066 | 4608/778 | 778 | 0.069 | 44.3%～51.54% ↑
> Mind2Web | 410K | 838 | 2553/654 | 654 | 0.045 | 4.06% ↑
>
> THREAD **consistently outperforms the Chunk paradigm across datasets of varying scales**, achieving higher element accuracy while maintaining acceptable costs. This demonstrates THREAD’s scalability in **handling both longer documents and larger corpora**.
>
> We hope this explanation clarifies THREAD’s scalability, and cost-effectiveness.

---

> ### Author Response · Authors · 2024-11-26
> **Kindly request your reply**
>
> Dear Reviewer yhzo,
>
> Thank you for your valuable feedback and constructive comments. We kindly ask if you could let us know whether our responses address your concerns and adjust your overall rating accordingly.
>
> If there are any remaining issues or clarifications needed, we would be happy to address them promptly.
>
> Sincerely,
>
> Authors of Submission 5754

---

### Official Review · Reviewer_4iT4 · 2024-11-04

**Soundness:** 3
**Presentation:** 1
**Contribution:** 3
**Rating:** 5
**Confidence:** 3

**Summary:**

1. The authors proposed to organize documents in Retrieval Augmented Generation systems into Logical Units, each consisting of Prerequisite, Header, Body, Linker, and Metadata. This aims to better answer how-to style questions.
2. To build a Threads knowledge base, LLMs are used to reformulate and extract logical units from a document corpus. To apply this in a RAG system, LU headers are used for embedding search, Prerequisite field is used for filtering, and Linker field is used for finding next steps in a dynamic style how-to questions when there can be multiple execution outcomes.
3. They tested this method on web navigation, wikihow, and human evaluation for incident mitigation, and showed superior performance when compared to other document chunking methods.

**Strengths:**

1. The authors proposed a novel way to organize document corpus. When compared to the usual chunking methods, it facilitates building logical connections between units, and compared to knowledge-graph approaches, it facilitates building meta connections between units that are larger than the more atomic-level knowledge base entities.
2. This approach is shown to be effective at improving answering how-to questions

**Weaknesses:**

### Clarity
In general the major problem for me is clarity. The paper is a bit hard to follow and requires multiple re-read to understand how the system is supposed to be implemented.

1. For the methodology section, reading the text itself is clearer than trying to parse figure 2 & 3. For figure 2, the arrow from (b) to the right side confused me. Per my understanding, the right panel is an individual view of an LU. Maybe split up the high-level flow with component details into different figures? In figure 3, it was initially hard to match the components to your text description. If you add some formulation and use symbols in both the figure and text, it might make it easier to follow.

2. For the experiment section, it is hard to establish correspondence between components described in the methodology section (sec 3) and their instantiations in the 3 datasets. e.g. I'm confused with how the "execution" part described in sec 3 is implemented in the 3 datasets.

- Having some symbols/anchors to establish correspondence between the subcomponents and their implementation in experiments might help.

- Maybe additional columns in table 1 showing design choices for each dataset like: "Dynamic?", "Executable?", "What is being executed" ...

- Before introducing the evaluation metrics, it might be helpful to explain what the task setup is for each dataset, what input is the baseline & your system given, what output is the system expected to produce.

### Some additional results that could be included
1. While the approach seems to perform well compared to existing benchmarks, I'm not really sold on the applicability of this approach. It requires rewrite & extraction of the documents using an LLM before it is even indexed. I would be interested to know how expensive time/cost-wise to implement this, e.g. how it scales in-terms of document length & corpus size etc.
2. The actual embedding search is using only the header field of each LU. It would be interesting to know if indexing using the body help or hurt performance.
3. Can you still use this organization approach for other 5W questions or would it hinder performance?

**Questions:**

My main concern is clarity, as it affects how I understand the presented results. I will be able to give a more accurate evaluation upon clarification.

### Methodology
1. How is the selector in the retrieval pipeline implemented? Does it check pre-requisites by embedding similarity or by string match/ngram overlap?
2. The Linker description says, “Its format varies by LU type, serving as either a query for retrieving other LUs or an entity relationship.” is it like an actual pointer to another LU or is it just some text that you then need to use the retriever to search for other LU?

### Experiments:
1. How do you verify the reformulation and extraction of documents to make sure there are no hallucinations?
2. For a single task in mind2web there are multiple interactions with the website to accomplish the task. Is this the task setup: given webpage and element choices, (optionally given relevant docs for RAG baselines), choose an element and an action?

### Results
In 5.2 ablation for Retrieval Unit Selector, it says "Comparing chunk selection to LU selection, the performance of Ele. Acc drops by 4.29% and Op.F1 by 3.38%", can you point out which rows on Table 5 is this concluded from? Is it a typo where "Chunk" performance is lower than "Chunk w/o chunk selection"?

---

> ### Author Response · Authors · 2024-11-23
> **Response to Reviewer 4iT4 [1/3]**
>
> ### **W1: Clarity**
> > ### C1:In figure 2, the arrow from (b) to the right side confused me... And in figure 3, it was initially hard to match the components to your text description... add some formulation and use symbols in both the figure and text.
>
> Thank you for your valuable feedback, which helps us improve the connection between our figures and textual descriptions.
>
> In Figure 2, the left part illustrates the construction process of THREAD: (a) represents the reformulation stage for structuring documents, and (b) depicts the process of extracting, merging, and constructing the final knowledge base. The right part provides a concise example of a document and an extracted logic unit. We remove the arrow and separate the two sections with dashed lines.
>
> Regarding Figure 3, it illustrates the workflow of RAG systems integrated with THREAD, with each component’s function explained in the caption. Following your suggestion, we incorporate symbols in both the figure and text to create stronger connections between them.
>
> We have revised Figures 2 and 3 in the updated version of our submission. We hope these improvements effectively address your concerns and make the explanations clearer.
>
> > ### C2:  Correspondence between components described in the methodology section (sec 3) and their instantiations in the 3 datasets. I'm confused with how the "execution" part described in sec 3 is implemented in the 3 datasets.
>
> Thanks for raising this concern, we are willing to clarify the components within LU for each dataset.
>
> The definitions of our logic units are consistent across all three datasets. For instance, the "Prerequisite" component includes supplementary information or actions that need to be executed beforehand. However, the content of the "Body" (execution) within the logic unit varies depending on the specific task. We have provided detailed descriptions of each dataset in Sections 4.1 and 4.2.
>
> To further clarification, below table lists the instantiations of the "Body" component across three datasets:
>
> Dataset | Body
> -------- | -----
> Mind2web | Website operation such as [CLICK, TYPE, SELECT, etc.] for specific HTML elements, along with an optional value.
> WikiHow |The step of completing certains tasks like inserting an image to powerpoint
> Incident Mitigation | Instructions, Command lines, Kusto queries, or Python code, etc
>
> Additionally, as noted on Line 308, we provide examples for each dataset in Appendix B, which helps to understand the task itself.
>
> We hope this clarification helps establish a clearer connection between the components in LU and their instantiations in each dataset.
>
> > ### C3: Maybe additional columns in table 1 showing design choices for each dataset.
>
> Thanks for your great suggestions, and we have added more details including the characteristics like 'Dynamic', 'Executable' of each dataset in Table 1 in our updated version.
>
> > ### C4: Before introducing the metrics, it might be helpful to explain what the task setup, input and outputs of baselines and your systems...
>
> In Section 4.3, we have introduced our baseline setting. The main baselines are RAG systems that rely on different data organization paradigms, and the inputs and outputs are the same as our systems.
>
> We have added the details about the input and output of each dataset in Section 4.4, and the details are as follows:
>
> - In Mind2Web (web navigation), the task is treated as a multi-choice question. At each step, the input includes HTML code, an instruction, and choices, while the output is the selected choice, and an operation from [CLICK, TYPE, SELECT, etc.] and an optional value to type in.
>
> - For Incident Mitigation and Wikipedia, the input consists of a question (which may include environmental details such as timestamps for reproduction), and the output corresponds to the action required at each step.
>
> We hope that the explanation addresses your concerns.

---

> ### Author Response · Authors · 2024-11-23
> **Response to Reviewer 4iT4 [2/3]**
>
> ### **W2: Additional results**
> > ### C1: The applicability of this approach. The time/cost-wise to implement, e.g. How it scales in-terms of document length & corpus size.
>
> Thank you for your thoughtful question regarding the scalability and cost-effectiveness of THREAD.
>
> - **Cost and performance analysis**
>
> We analyze the preprocessing overhead for constructing the final knowledge base using the Mind2Web dataset and compare THREAD to other paradigms that also leverage LLMs during preprocessing. The results are summarized as follows:
>
> Paradigm | Chat tokens (I/O) / Doc  | Embedding tokens / Doc | Price / Doc ($) |Chat Model API Call times / Doc | Ele. Acc
> -------- | -----  | ----- | ----- | ----- | -----
> Recursive (Chunk) | / | 838 |0.000084 | / | 64.23
> Semantic | / | 3802 | 0.00038 | / | 65.14
> Proposition | 61836/388 | 978 | 0.63 | 28.8 | 62.37
> THREAD | 2553/654 | 654 | 0.045 | 1 | 68.29
>
> From this table, THREAD **achieves the highest element accuracy (68.29%) with a balanced trade-off between cost and performance**. While its cost is higher than the Chunk and Semantic paradigms, it is significantly lower than the Proposition paradigm, making it suitable for real-world applications. Additionally, in Section 5.4, we show that Llama3-70B achieves comparable performance with GPT-4, further emphasizing THREAD’s effectiveness and generability.
>
> - **Scalability to Larger Datasets**
>
> More importantly, there are currently no open-source datasets available for evaluating the performance on how-to questions, making data collection a particularly challenging task.
>
> To evaluate THREAD’s scalability, we have curated datasets by transforming existing datasets (e.g., Mind2Web) and collecting data from the internet (e.g., WikiHow) and examined its performance on three datasets with different sizes. The table below summarizes the results:
>
> Dataset | Total Tokens Of Dataset  | Original Tokens / Doc | Chat tokens (I/O) / Doc  | Embedding tokens / Doc | Price / Doc ($) | Ele. Acc Improvement (THREAD v.s. Chunk)
> ---- | ---- | ---- | ---- | ---- | ---- | ---- |
> Incident Mitigation | 112K  | 2002 | 2814/1464 | 1464 | 0.072 | 23.68%～36.68% ↑
> WikiHow | 200K | 2066 | 4608/778 | 778 | 0.069 | 44.3%～51.54% ↑
> Mind2Web | 410K | 838 | 2553/654 | 654 | 0.045 | 4.06% ↑
>
> THREAD **consistently outperforms the Chunk paradigm across datasets of varying scales**, achieving higher element accuracy while maintaining acceptable costs. This demonstrates THREAD’s scalability in **handling both longer documents and larger corpora**.
>
> We hope this explanation clarifies THREAD’s scalability, and cost-effectiveness.
>
> > ### C2: Embedding search uses only the header. How about using the body.
>
> Thank you for your question.
>
> We have indeed considered this point but decided to follow the approach proposed by LlamaIndex (2023), which demonstrates that *summary-based indexing improves retrieval accuracy and efficiency compared to indexing the full text*, while still enabling comprehensive responses by accessing the original content. In our method, we use the 'Header' for indexing rather than the 'Body'.
>
> As explained in Lines 161–192 of our paper, the 'Body' provides detailed steps or actions, whereas the 'Header' summarizes the intent of the LU, making it more effective for retrieval purposes. This design ensures that our system efficiently retrieves relevant LUs.
>
> [1] J. Liu. A new document summary index for llm-powered qa systems. https://www.llamaindex.ai/blog/a-new-document-summary-index-for-llm-powered-qa-systems-9a32ece2f9ec, 2023.
>
> > ### C3: Can you still use this organization approach for other 5W questions or would it hinder performance?
>
> Thank you for your question.
>
> To evaluate whether our organization paradigm can handle '5W' questions without compromising performance, we conduct experiments on **2WikiMultiHopQA**. Specifically, we sample 200 questions from two types, inference and compositional type, like "Who is the maternal grandfather of Abraham Lincoln?" and "Who is the founder of the company that distributed La La Land film?" During our experiment, we use the relations between characters or things as the 'linker'.
>
> The results evaluated using Exact Match, F1 score, and accuracy, are shown below:
>
> Method | F1 | EM | Acc
> ---- | ---- | ---- | ---- |
> Chunk | 24.23 | 15.50 | 25.00
> THEAD | **44.77** | **30.00** | **46.00**
>
> These findings demonstrate that THRED not only surpass the chunk-based paradigm on multi-hop questions, but also excels at handling more complex how-to question.
>
> [1] Ho X, Nguyen A K D, Sugawara S, et al. Constructing a multi-hop QA dataset for comprehensive evaluation of reasoning steps[J]. arXiv preprint arXiv:2011.01060, 2020.

---

> ### Author Response · Authors · 2024-11-23
> **Response to Reviewer 4iT4 [3/3]**
>
> ### **Q1: Methodology**
> > ### C1: How is the selector in the retrieval pipeline implemented? Does it check pre-requisites by embedding similarity or by string match/ngram overlap?
>
> We first utilize text-embedding similarity to identify the top-5 candidate LUs. Then, we employ an LLM to select the most suitable LU from these candidates. In this way, it enables the LLM to verify whether the prerequisites are met based on the context and history while reducing bias from relying solely on similarity scores.
>
> The prompt used for the selector is provided in Appendix 3.1, titled "Instruction that selects the most relevant logic unit based on user query and chat history." And we also conduct an ablation study, as shown in Table 5, to demonstrate the effectiveness of the selector.
>
> > ### C2: Is the linker like an actual pointer to another LU or is it just some text that you then need to use the retriever to search for other LU?
>
> The linker works like a pointer in text format which points out the intention of the next step based on different execution results. In each step, it requires the module "Query Generation" in Figure 3 to correctly map the query for the next retrieval.
>
> ### **Q2: Experiments**
> > ### C1: How to verify the reformulation and extraction of documents to make sure there are no hallucinations?
>
> Thank you for your thoughtful question. To ensure the reformulation and extraction processes are free from hallucinations, we adapt the following strategies:
>
> 1. **LLM excels at parsing structured information from docs**
>
> While hallucinations are more common in reasoning or inference tasks, research [1] has shown that LLMs excel at information extraction tasks, such as parsing structured data from documents. Since manually reformulating unstructured documents into a structured format is labor-intensive, leveraging LLMs is a practical and efficient solution.
>
> 2. **Two-Stage Refinement Process**
>
> To further minimize hallucinations, we implement a two-stage approach. After reformulation, the original document and its reformulated version are provided to the LLM for refinement. This iterative process ensures higher fidelity and reduces the likelihood of errors in the structured ones.
>
> These measures help maintain the accuracy and reliability of the reformulation and extraction pipeline.
>
> [1] Xu D, Chen W, Peng W, et al. Large language models for generative information extraction: A survey[J]. Frontiers of Computer Science, 2024, 18(6): 186357.
>
>
> > ### C2: Is this the task setup of Mind2web: given webpage and element choices, (optionally given relevant docs for RAG baselines), choose an element and an action?
>
> Yes, as detailed in W1-C4, we adhere to the input format and evaluation settings described in [1]. Specifically, the input includes HTML code (as provided by the dataset), instructions, and choices. For RAG-based methods, we additionally provide the relevant documents.
>
> [1] Deng X, Gu Y, Zheng B, et al. Mind2web: Towards a generalist agent for the web[J]. Advances in Neural Information Processing Systems, 2024, 36.
>
> ### **Q3: Results**
> > ### C1: Is it a typo where "Chunk" performance is lower than "Chunk w/o chunk selection"?
>
> Thanks for pointing out our typo error. Here we reveal that by utlizing the chunk selection, the performance of Chunk-based method drops across each metrics. But for Thread, we consistently improve the performance on each metric.

---

> ### Comment · Reviewer_4iT4 · 2024-11-25
>
> Thank you for taking the time to clarify some of the details.
> However, I am still not sold on the applicability of this method in real-world scenario.
> For example, in your **Cost and performance analysis** and **Scalability** response, engineering hours are not taken into account:
>
> 1.  adapting LLM reformulation to each new task (WebNav/WikiHow/Incident Mitigation ... ) requires crafting a new set of prompt
>
> 2. verifying reformulation quality is still non trivial,
>     * > research [1] has shown that LLMs excel at information extraction tasks
>
>     rewriting entire document is not the same as extracting named entities/semantic triples/events (which is more like sentence level stuff) in information extraction
>     * While I think the **Two-Stage Refinement Process** intuitively makes sense,
>        > This iterative process ensures higher fidelity and reduces the likelihood of errors in the structured ones.
>
>        this claim is not supported with results in your paper.
>
> Additionally, re your response to **C2: Embedding search uses only the header. How about using the body.**, where
> >  we decided to follow the approach proposed by LlamaIndex (2023), which demonstrates that summary-based indexing improves retrieval accuracy and efficiency compared to indexing the full text
>
> The reference you provided does not have experimental results to support this statement, maybe consider provide results on one of the datasets in your paper to back this claim?

---

> > ### Author Response · Authors · 2024-11-25
> >
> > Dear reviewer,
> >
> > Thank you for your feedback. Regarding the questions you raised, we would like to clarify them accordingly.
> >
> > > ### C1: I am still not sold on the applicability of this method in real-world scenarios.
> >
> > We have conducted experiments across three real-world scenarios, including two public settings, Web Navigation and WikiHow Instructions, and an industrial setting, Incident Mitigation. Each of these scenarios focuses on addressing real-world how-to questions. As shown in Table 3, THREAD outerforms the chunk-based paradigm by 23.68%~36.68% in Success Rate on IcM, demonstrating its potential applicability to other real-world scenarios.
> >
> > > ### C1.1: adapting LLM reformulation to each new task (WebNav/WikiHow/Incident Mitigation ... ) requires crafting a new set of prompt
> >
> > Leveraging LLMs to process documentation is one of the best ways compared with manual processing considering the scalability and cost (as detailed in W2-C1). Comparing with other ML-based methods, prompting LLMs gives a far much cheaper general, and flexible way.
> >
> > Moreover, prompt engineering in our document reformulation task is general. This task is inherently task-agnostic, for example, in Appendix 3.1, we provide a reformulation prompt where most of the content consists of the definitions of components (Lines 744–750) and guidelines for reformulation (Lines 867–899). Adapting this to new scenarios requires only minor changes, such as updating placeholders like <TSG> (Line 740), brief descriptions of original documents, and two manually reformulated examples. This approach reduces engineering overhead and aligns with scalable solutions for new scenarios.
> >
> > We hope this clarification demonstrates the flexibility and scalability of our method. And the engineering hours are unnecessary and unpredictable for cost analysis.
> >
> > > ### C1.2. verifying reformulation quality is still non trivial
> >
> > > Comment: rewriting entire document is not the same as extracting named entities/semantic triples/events in information extraction.
> >
> > We would like to clarify that our operation is 'reformulation', not 'rewriting'. It is closer as a reorganization task with few context changes.. As illustrated in Table 9, the "### Linker" section in the reformulated document (Lines 833–838) is derived directly from the context of the original document (Lines 819–824). This process does not change the original description, but instead parses and reorganizes the information explicitly, making it easier for RAG systems to process.
> >
> > Additionally, it is a common practice that LLM excel at document restruction tasks, for example, these two research [1,2] has demonstrated that LLMs excel at transforming unstructured or semi-structured information into structured formats. These findings ensure the reliability of our reformulation process.
> >
> > [1] Ko H, Yang H, Han S, et al. Filling in the gaps: Llm-based structured data generation from semi-structured scientific data[C]//ICML 2024 AI for Science Workshop. 2024.
> >
> > [2] Zhong A, Mo D, Liu G, et al. LogParser-LLM: Advancing Efficient Log Parsing with Large Language Models[C]//Proceedings of the 30th ACM SIGKDD Conference on Knowledge Discovery and Data Mining. 2024: 4559-4570.
> >
> > > Comment: While I think the Two-Stage Refinement Process intuitively makes sense, this claim is not supported with results in your paper.
> >
> > The refinement is a stage leveraging LLM-as-a-judge paradigm, which is introduced to reduce hallucinations and prevent overlooking details. As described in our paper, both the original document and its reformulated version are provided to the LLM to facilitate a comprehensive refinement.
> >
> > The LLM-as-a-judge paradigm has gained significant traction in the LLM community, as it empowers LLMs to critically evaluate, refine, and improve outputs. For example, Zheng et al. [1] demonstrated that LLMs could reliably act as evaluators, improving response quality and consistency. Similarly, Huang et al. [2] highlighted the utility of fine-tuned judge models as task-specific classifiers, further substantiating the efficacy of LLMs in evaluation and refinement tasks.
> >
> > While the effectiveness of this approach is well-supported by existing research, our focus lies in applying it to construct our knowledge base, rather than independently validating the paradigm itself. Therefore, the refinement process is introduced as a beneficial step but is not the focus of this paper.
> >
> > [1] Zheng L, Chiang W L, Sheng Y, et al. Judging llm-as-a-judge with mt-bench and chatbot arena[J]. Advances in Neural Information Processing Systems, 2023, 36: 46595-46623.
> >
> > [2] Huang H, Qu Y, Liu J, et al. An empirical study of llm-as-a-judge for llm evaluation: Fine-tuned judge models are task-specific classifiers[J]. arXiv preprint arXiv:2403.02839, 2024.

---

> > > ### Author Response · Authors · 2024-11-25
> > >
> > > > ### C2: Embedding search uses only the header. How about using the body... The reference you provided does not have experimental results to support this statement
> > >
> > > While LlamaIndex suggests that summary-based indexing improves retrieval accuracy and efficiency compared to full-text indexing, this conclusion is further supported by research [2, 3]. Specifically, Figure 7 in [1] and the figure captioned "Combining Contextual Embedding and Contextual BM25 reduces the top-20-chunk retrieval failure rate by 49%." in [2] demonstrate that utilizing summarization or contextual information significantly enhances retrieval accuracy.
> > >
> > > These findings validate our approach of using the 'Header' instead of the 'Body' for LU retrieval, reinforcing the effectiveness of summary-based indexing for this purpose.
> > >
> > > To further explore this point, we conduct additional experiments to evaluate the impact of indexing by the 'Body.' Using a subset of data (~15%) sampled from Mind2Web (indicated by *), the results are as follows:
> > >
> > > Method | Ele. Acc | Op. F1 | Step SR
> > > --- | --- | --- | ---
> > > THEAD w. Header | 68.29   | 69.53 | 61.94
> > > THEAD w. Body* | 66.28 | 68.48 | 56.89
> > > **$\Delta\downarrow$** | **2.03** |  **1.05** | **5.05**
> > >
> > > The results clearly show that indexing by the 'Body' leads to a decline in performance across all metrics, particularly in Step Success Rate, where the drop is substantial. This demonstrates the superiority of using the 'Header' for indexing in THREAD and further validates our design choice.
> > >
> > > [1] J. Liu. A new document summary index for llm-powered qa systems. https://www.llamaindex.ai/blog/a-new-document-summary-index-for-llm-powered-qa-systems-9a32ece2f9ec, 2023.
> > >
> > > [2] Eibich M, Nagpal S, Fred-Ojala A. ARAGOG: Advanced RAG Output Grading[J]. arXiv preprint arXiv:2404.01037, 2024.
> > >
> > > [3] Introducing Contextual Retrieval. https://www.anthropic.com/news/contextual-retrieval, 2024.
> > >
> > >
> > >
> > > We would appreciate it greatly if you could let us know whether our responses address your concerns and adjust your rating accordingly.
> > >
> > > Sincerely,
> > >
> > > Authors of Submission 5754

---

> > > > ### Comment · Reviewer_4iT4 · 2024-11-25
> > > >
> > > > Just to clarify, are you feeding header+body or body-only for this result?
> > > > > THEAD w. Body*

---

> > > > > ### Author Response · Authors · 2024-11-26
> > > > >
> > > > > Thanks for your question, as you raised in W2-C2, we only use the Body for indexing.

---

> > > > > > ### Comment · Reviewer_4iT4 · 2024-11-26
> > > > > >
> > > > > > Apologies for the confusion. I meant whether header+body (hence “adding body”) would hurt or improve

---

> > > > > > > ### Author Response · Authors · 2024-11-26
> > > > > > >
> > > > > > > No worries. As outlined in our latest response, research from Anthropic [3] validates this point. By providing a concise context for a single chunk and concatenating it with the original chunk for retrieval, the figure captioned "Combining Contextual Embedding and Contextual BM25 reduces the top-20-chunk retrieval failure rate by 49%" in [2] demonstrates that utilizing contextual information significantly enhances retrieval accuracy. This approach aligns with indexing using 'header+body.'
> > > > > > >
> > > > > > > However, whether we index by 'header' or 'header+body' is not the primary focus of our work and does not affect our main conclusions. Our experiments that index by 'header' just demonstrate the effectiveness of THREAD.
> > > > > > >
> > > > > > > We hope this clarifies your concerns.
> > > > > > >
> > > > > > > [3] Introducing Contextual Retrieval. https://www.anthropic.com/news/contextual-retrieval , 2024.

---

> ### Author Response · Authors · 2024-11-26
> **Kindly request your reconsideration**
>
> Dear Reviewer 4iT4,
>
> We kindly request that you reconsider your score in light of the clarifications and revisions we have made, and we also welcome any further discussions you may have. Thank you once again for your thoughtful feedback and for contributing to the improvement of our work.
>
>
> Sincerely,
>
> Authors of Submission 5754

---

### Author Response · Authors · 2024-12-03
**General Responses to All Reviewers**

We would like to thank all the reviewers for their time and effort in the review process. We appreciate that you found our work as "novel, original, and reasonable" [`4iT4`, `yhzo`, `pBXn`], "well-written and clear" [`oQG8`, `yhzo`, `pBXn`], our focus on how-to questions as "good and important" [`oQG8`, `pBXn`], and our experiments as "extensive and effective" [`4iT4`, `yhzo`, `pBXn`]. Taking into account all the reviewers' comments, we've responded to each reviewer individually and uploaded a revised manuscript with changes marked in blue.

Following the reviewers' suggestions, the key revisions we made are as follows:

- **Preprocessing overhead analysis**: We added an analysis of the preprocessing overhead involved in constructing the final knowledge base, comparing THREAD to other data organization paradigms, in Appendix E.1 [`4iT4`, `yhzo`].

- **Scalability validation**: To demonstrate THREAD’s scalability across different real-world scenarios, we included statistical data on the scale, cost, and performance improvements of each dataset in Appendix E.1 [`4iT4`, `yhzo`, `oQG8`].

- **Generalization to factoid questions**: We added experimental results on multi-hop QA (2WikiMultiHopQA) in Appendix E.2 to explore THREAD's generalization [`4iT4`].

- **Improved figure and table clarity**: We revised the content and captions of Figure 2 and Figure 3 for clarity and added two new columns in Table 1 to describe the characteristics of each dataset [`4iT4`].

- **Details on LU merging**: We provided additional details about merging logic units with similar headers in Appendix C.2 [`pBXn`].

Please kindly check our updated submission. Once again, we deeply appreciate your valuable feedback and time.

---

### Meta-Review · Area_Chair_SVXP · 2024-12-23

**Metareview:**

This paper presents a way to segment sources in RAG systems and focuses on a particular type of questions and show strong results in end to end experiments.  The paper has a lot of experiments and the motivation is clear.  That said, I am not particularly excited about the paper due to a few reasons:

1.  Focus on a relatively narrow type of questions.
2.  A fairly specific way to segment documents (which the second reviewer points out as well) and I am uncertain whether this will generalize to a variety of sources.

Given this I think this paper will be more well suited as a tech report and not a paper at ICLR.  Most reviewers also think similarly and have a lukewarm response to the paper.

**Additional Comments On Reviewer Discussion:**

There is a bunch of discussion between reviewers and authors for this paper.

---

### Decision · Program_Chairs · 2025-01-22

Reject